# VidLaDA: Bidirectional Diffusion Large Language Models for Efficient Video Understanding

**Zhihao He** [1 2]  **Tieyuan Chen** [1 3]  **Kangyu Wang** [1 2]  **Ziran Qin** [1]  **Yang Shao** [4]  **Chaofan Gan** [1]  **Shijie Li** [1]
**Zuxuan Wu** [† 5 2]  **Weiyao Lin** [† 1]

## Abstract

Current Video Large Language Models (Video LLMs) typically encode frames via a vision encoder and employ an autoregressive (AR) LLM for understanding and generation. However, this AR paradigm inevitably faces a dual efficiency bottleneck: strictly unidirectional attention compromises *understanding efficiency* by hindering global spatiotemporal aggregation, while serial decoding restricts *generation efficiency*. To address this, we propose **VidLaDA**, a Video LLM based on Diffusion Language Models (DLMs) that leverages bidirectional attention to unlock comprehensive spatiotemporal modeling and decode tokens in parallel. To further mitigate the computational overhead of diffusion decoding, we introduce **MARS-Cache**, an acceleration strategy that prunes redundancy by combining asynchronous visual cache refreshing with frame-wise chunk attention. Experiments show VidLaDA rivals state-of-the-art AR baselines (e.g., Qwen2.5-VL and LLaVA-Video) and outperforms DLM baselines, with MARS-Cache delivering over $12\times$ speedup without compromising accuracy. Code and checkpoints are open-sourced at https://github.com/ziHoHe/VidLaDA.

## 1. Introduction

Video Large Language Models (Video LLMs) (Lin et al., 2024; Cheng et al., 2024) have recently emerged as a key paradigm for connecting visual perception with language reasoning, showing strong potential in tasks like video captioning, spatiotemporal question answering, and long-horizon decision making. Most existing systems follow a standard recipe that couples a pretrained vision encoder (e.g., ViT (Zhu et al., 2023; Tschannen et al., 2025)) with an Autoregressive (AR) LLM (Grattafiori et al., 2024; Team et al., 2024). Video frames are first encoded into visual embeddings and projected into the language embedding space, after which the model generates responses via next-token prediction under a causal attention mask (Lin et al., 2024).

While this AR-based paradigm has driven rapid progress, it potentially ignores the fundamental mismatch between AR and the spatiotemporal nature of video. Unlike text, visual semantics (objects, relations, and event cues) are distributed across space and time without an inherent left-to-right ordering (Yu & Wang, 2025). However, after rasterizing video tokens into a 1D sequence and feeding them into an AR decoder, the causal mask enforces a strictly unidirectional dependency. This creates an asymmetric receptive field where early tokens are visible to many subsequent positions while late tokens are structurally under-attended, inducing positional bias and non-uniform utilization of visual evidence. This results in suboptimal understanding efficiency, i.e., the Video LLM fails to fully exploit the available bidirectional visual information within its receptive field. We formalize this inefficiency issue from two complementary perspectives: (i) causal attention yields a visibility-frequency imbalance that encourages *early receptive field* behaviors (Proposition 3.1); and (ii) due to the restricted access to future visual tokens, the AR decoding stack admits a strictly *lower upper bound* on the usable information extracted from the bidirectional vision encoder (Proposition 3.2). Empirically, these biases manifest in two ways: brittle intra-frame performance under controlled relocation of high-information patches, and degraded temporal robustness in causal video question answering when key evidence is sparse or occurs at different positions in the video (see Figure 2).

Moreover, video understanding is increasingly shifting from short-horizon perception to complex spatiotemporal reasoning (Ye et al., 2025b; Liu et al., 2025; Feng et al., 2025). Compared to images, videos are both temporally redundant and dynamically evolving, which often requires models to

---

†Corresponding authors.  ¹Shanghai Jiao Tong University ²Shanghai Innovation Institute ³ZhongguanCun Academy ⁴Shenzhen International Graduate School, Tsinghua University ⁵Fudan University. Correspondence to: Zhihao He <ziho_he@sjtu.edu.cn>, Weiyao Lin <wylin@sjtu.edu.cn>.

*Proceedings of the 43rd International Conference on Machine Learning*, Seoul, South Korea. PMLR 306, 2026. Copyright 2026 by the author(s).

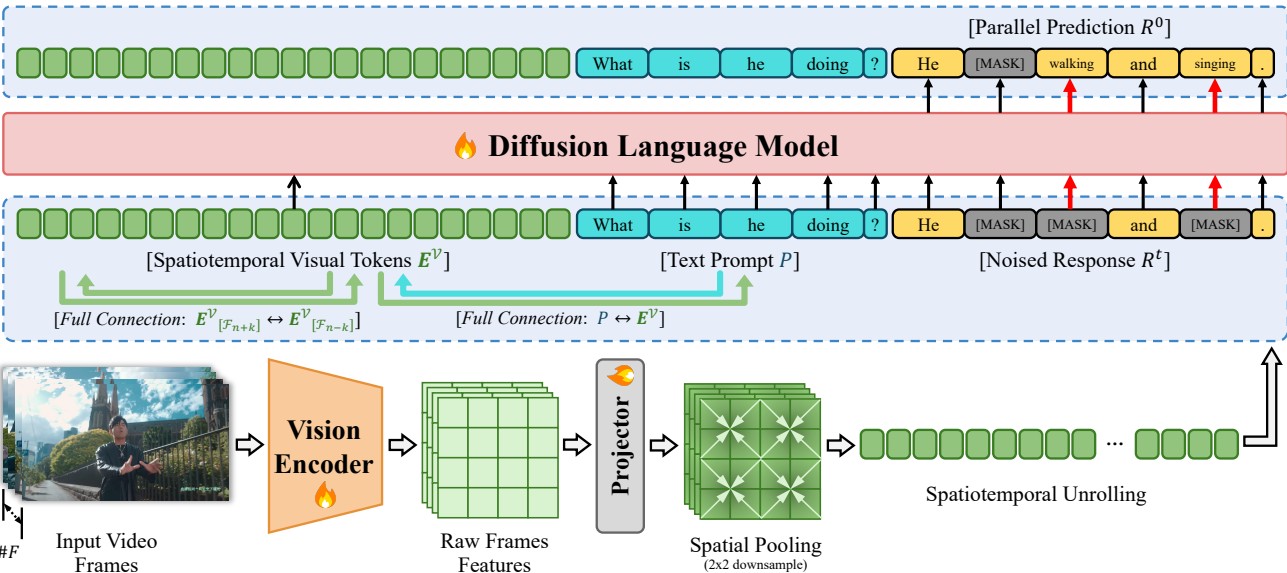

**Figure 1.** **The overall architecture of VidLaDA.** Input video frames $\mathcal{V}$ are encoded and spatially pooled (via $2 \times 2$ downsampling) before being unrolled into a sequence of Spatiotemporal Visual Tokens $\boldsymbol{E}^{\mathcal{V}}$. These tokens, combined with the text prompt $P$ and the noised response $R^t$, are processed by the **Diffusion Language Model**. Unlike autoregressive models, VidLaDA utilizes full **bidirectional attention**. This design enables global, unconstrained interactions both within and across visual and textual modalities, ultimately facilitating the parallel prediction of the target response $R^0$.

efficiently aggregate and understand key spatiotemporal evidence across the video and form a long-form reasoning chain. However, the standard AR paradigm faces a dual efficiency bottleneck in this context. First, regarding *understanding efficiency* as analyzed above, strictly unidirectional temporal modeling can hinder capturing global temporal relationships across different moments in a video (Guo et al., 2025b; Chen et al., 2024b), which limits the effective aggregation of spatiotemporal evidence. Second, regarding *generation efficiency*, AR decoding is inherently serial, making latency scale linearly with generated tokens and limiting generation efficiency for long-form reasoning for complex spatiotemporal reasoning. Conversely, Diffusion Language Models (DLMs) offer a compelling alternative: generation is formulated as iterative denoising in a discrete space with full bidirectional attention (Nie et al., 2025), which removes the causal constraint and enables parallel prediction of multiple tokens per step (Wu et al., 2025). This bidirectional decoding is advantageous for video understanding, where key spatiotemporal evidence should aggregate visual context globally across all frames and patches rather than follow a forced causal path.

Despite these advantages, directly applying DLMs to long-form video introduces a new efficiency bottleneck. Unlike AR inference, where prefix key/value states can be cached, and only the newest token is processed, vanilla DLM inference recomputes bidirectional attention over the entire multimodal sequence at every decoding step. With thousands of video tokens $\boldsymbol{E}^{\mathcal{V}}$, this results in prohibitive $\mathcal{O}(|\boldsymbol{E}^{\mathcal{V}}|^2)$

attention cost per step, which can negate the speed benefits of parallel decoding. Moreover, generic DLM acceleration techniques developed for text-only settings (Wu et al., 2025) ignore multimodal heterogeneity: in our analysis, visual and textual states exhibit markedly different decoding dynamics, and inter-frame attention presents strong local temporal structure with a small set of globally influential anchor tokens. Treating all modalities and all layers identically, therefore wastes computation on repeatedly recomputing stable context, especially for video inputs.

To unlock bidirectional diffusion for efficient video understanding, we propose **VidLaDA** (**Vid**eo-**La**nguage **D**iffusion with m**A**sking), a Video LLM built on the DLM backbone with bidirectional attention to maximize *understanding efficiency*, and **MARS-Cache** (**M**ulti-modal **A**synchronous **R**efreshing **S**trategy), an inference framework that prunes redundancy specific to multimodal diffusion to maximize *generation efficiency*. Trained via a multi-stage curriculum on a comprehensive video dataset incorporating our newly constructed collection, VidLaDA couples a strong vision encoder with DLM-based bidirectional decoding, enabling unconstrained global interactions across video tokens, prompts, and partially denoised responses. MARS-Cache accelerates DLM-based Video LLM decoding by combining (i) frame-wise chunk attention to exploit temporal locality, reducing the visual refresh cost to linear $\mathcal{O}(|\boldsymbol{E}^{\mathcal{V}}|)$, (ii) adaptive anchor token searching to preserve necessary global connectivity, and (iii) modality- and depth-wise asynchronous cache refreshing to update

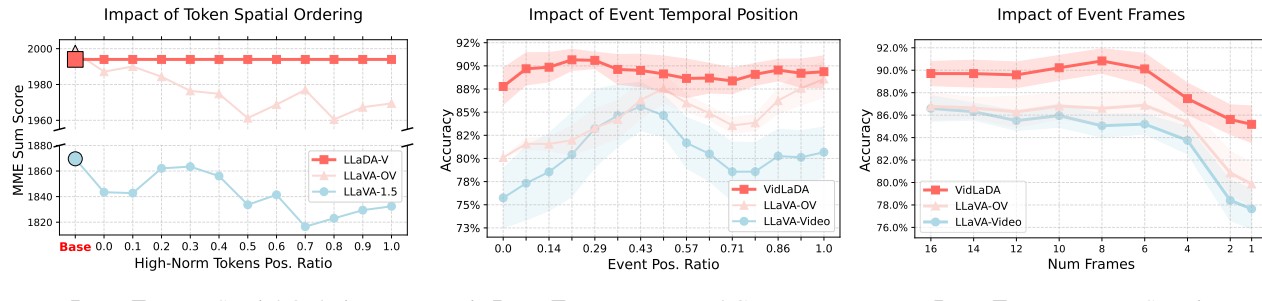

*(a)* **Intra-Frame:** Spatial Ordering.   *(b)* **Inter-Frame:** Temporal Context.   *(c)* **Inter-Frame:** Frame Sparsity.

*Figure 2.* **Comparison of Spatiotemporal Robustness. (a)** Performance vs. spatial location of high-norm tokens. DLM baseline remains invariant, whereas AR degrades when salient features shift from the start. **(b)** Performance vs. temporal location of the key event. DLM baseline demonstrates stability across the timeline, while AR baselines show significant volatility. **(c)** DLM baseline maintains high accuracy with fewer frames, demonstrating superior aggregation of sparse evidence compared to AR models.

stable visual context less frequently than dynamic textual states. Our main contributions are summarized as follows:

1. We introduce VidLaDA, the first family of DLM-based Video LLMs. By utilizing bidirectional decoding, Vid-LaDA mitigates the *asymmetric receptive field* issue in causal attention and *improves the theoretical upper bound* of spatiotemporal understanding in Video LLMs.

2. VidLaDA consistently outperforms existing DLM-based baselines (e.g., LLaDA-V (You et al., 2025) and Dream-VL (Ye et al., 2025a)). Furthermore, it remains highly competitive with state-of-the-art open-sourced AR-based Video LLMs (e.g., Qwen2.5-VL (Bai et al., 2025b) and LLaVA-Video (Zhang et al., 2024b)).

3. We propose MARS-Cache, a Multi-modal Asynchronous Refreshing Strategy, motivated by observations of modality-wise stability and attention structure during decoding. Substantially, MARS-Cache achieves a more than $12\times$ throughput improvement compared to vanilla DLM without compromising reasoning accuracy.

## 2. Preliminary

**Problem Formulation.** Given a video input $\mathcal{V}$ and a textual prompt $P$, the goal of a Video Large Language Model is to generate a target response $R = \{R_1, \ldots, R_{|R|}\}$. The video is typically processed into visual tokens $\boldsymbol{E}^{\mathcal{V}}$, which form the multimodal context alongside prompt textual tokens $P$.

**From Autoregressive to Diffusion Modeling.** Standard Video LLMs adopt an AR paradigm (Li et al., 2024a; Zhang et al., 2024b). They model the joint probability of the response as a product of conditional probabilities, optimized via the negative log-likelihood loss. Crucially, AR models rely on the *causal attention*, where the token can only attend to its preceding ones. This unidirectional dependency inherently prevents early tokens (visual or textual) from interacting with subsequent context, limiting the model's

ability to capture global spatiotemporal relationships.

In contrast, our VidLaDA is built upon Diffusion Language Models (DLMs) (Nie et al., 2025; Ye et al., 2025c), which treat generation as a bidirectional iterative denoising process. Instead of sequential prediction, DLMs utilize a Masked Diffusion Model (MDM) framework. During training, a subset of response tokens is randomly replaced by a special [MASK] token based on a timestep $t$. The model learns to predict the original identities of these masked tokens simultaneously (Nie et al., 2025). Moreover, the DLM architecture allows VidLaDA to employ a *full bidirectional attention* mechanism without causal masking. Consequently, the aggregation and prediction of any token depends on the entire global context (both unmasked and masked tokens in $\boldsymbol{E}^{\mathcal{V}}$, $P$, and $R$), facilitating unconstrained interaction between visual and textual modalities. For detailed mathematical formulations of the forward and reverse diffusion processes, please refer to Appendix B.

## 3. VidLaDA: Efficient Video Understanding

### 3.1. Why Bidirectional DLM for Video Understanding?

**Proposition 3.1.** *AR possesses an asymmetric receptive field that precludes uniform spatiotemporal processing.*

*Proof.* See Appendix D.1. □

**Proposition 3.2.** *Subject to non-degenerate encoding assumptions, the AR paradigm inherently limits the cumulative pointwise information capacity, resulting in a strictly lower theoretical upper bound of spatiotemporal understanding compared to a bidirectional decoding paradigm:*

$$\sum_{t=1}^{T} \sup I(\boldsymbol{z}_t^{bidi}; \mathcal{V}) > \sum_{t=1}^{T} \sup I(\boldsymbol{z}_t^{AR}; \mathcal{V}), \quad (1)$$

*where $\mathcal{V}$ denotes the original visual inputs, and $\boldsymbol{z}_t^*$ denotes the representation encoded by ViT and LLM at timestep $t$.*

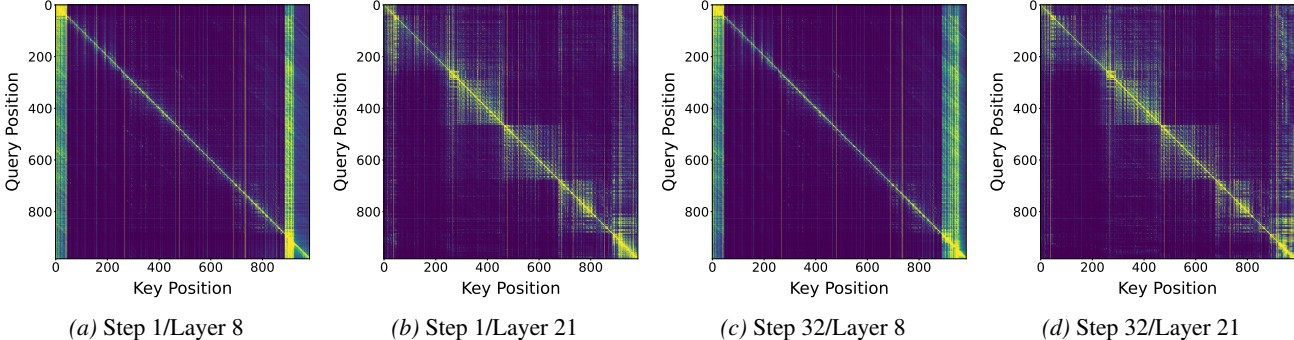

|  |  |  |  |
|---|---|---|---|
| *(a)* Step 1/Layer 8 | *(b)* Step 1/Layer 21 | *(c)* Step 32/Layer 8 | *(d)* Step 32/Layer 21 |

*Figure 3.* **Visualization of Attention Patterns.** We display the attention score matrices across different decoding steps and layers. The heatmaps reveal two distinct structural properties utilized by our MARS-Cache design: (1) Chunk-wise Locality, visible as diagonal blocks where tokens primarily attend to their temporal neighbors, and (2) Global Anchor Tokens, manifested as prominent vertical bands where specific tokens consistently attract global attention from the entire sequence, regardless of the diffusion step or network depth.

*Proof.* See Appendix D.2. □

### 3.1.1. THE IMPACT OF TOKEN SPATIAL ORDERING

To empirically validate the asymmetry described in Proposition 3.1, we investigate how the spatial ordering of visual tokens affects semantic stability at the single-frame level.

In ViT, the patch order is conventionally rasterized (top-left to bottom-right). Inspired by (Luo et al., 2025), we use the $\ell_2$-norm of the ViT output features as a proxy for the information density of visual tokens. We identify the top-$k$ high-norm tokens and virtually relocate them within the sequence fed to the LLM, but maintaining the original positional IDs in sequence. This design isolates the effect of causal visibility in the decoder from changes in spatial positional encoding. We define a position ratio $r \in [0, 1]$, where $r = 0$ places high-norm tokens at the start of the visual sequence and $r = 1$ pushes them to the end.

As illustrated in Figure 2a, we evaluate the models on the MME (Fu et al., 2026) benchmark under this shuffling protocol (cf. Appendix G.2). The bidirectional DLM baseline (You et al., 2025) exhibits a near-constant performance profile (red line), demonstrating invariance to the causal position of semantic features.

Conversely, AR baselines (LLaVA-OneVision (Li et al., 2024a), LLaVA-1.5 (Liu et al., 2024b)) display performance degradation, particularly when high-information tokens are shifted towards the middle or end of the sequence. This corroborates Proposition 3.1: AR models over-rely on early tokens as a receptive field to maintain semantic interpretation. When salient features are displaced from the high-visibility start positions to later positions (where visibility is lower), the causal attention mechanism fails to allocate sufficient attention mass to them, treating them as contextually less significant solely due to their temporal index.

### 3.1.2. THE IMPACT OF EVENT POSITION AND SPARSITY

To verify spatiotemporal robustness in realistic settings beyond artificial single-image shuffling, we evaluate on ReX-Time (Chen et al., 2024a) (cf. Appendix G.3), where each question depends on a specific event segment. We bucket samples by the temporal position ratio of the ground-truth event and report accuracy under uniform frame sampling.

As shown in Figure 2b, AR baselines exhibit a U-shaped sensitivity to event position. Since the original spatiotemporal structure is preserved (unlike the single-frame experiment), this trend reflects the interplay between causal masking and positional effects such as RoPE long-range attenuation (Su et al., 2024) and recency bias (Liu et al., 2024c). Specifically, events occurring early suffer from long-range attenuation under RoPE, while mid-sequence events ($40\% \sim 80\%$) are particularly fragile: they are neither close enough to benefit from recency nor sufficiently early to act as the early receptive field under causal decoding (Proposition 3.1), making their evidence more likely to be diluted or overwritten during autoregressive abstraction.

Notably, LLaVA-Video (Zhang et al., 2024b) exhibits the most severe instability despite extensive video training compared to LLaVA-OneVision (Li et al., 2024a), suggesting that scaling data alone cannot remove the structural limitations imposed by the early receptive field of AR. This limitation becomes more evident when event evidence is sparse. As shown in Figure 2c, AR models degrade more sharply as fewer event frames are available, indicating that answer-relevant information of the event is more likely to be dispersed during deep semantic abstraction once it is not continuously reinforced under causal attention. In contrast, the bidirectional DLM baseline maintains a near-flat accuracy profile across event positions (Figure 2b) and remains accurate even under sparse event frames (Figure 2c). This behavior is consistent with Proposition 3.2: bidirectional decoding allows answer-relevant representations to

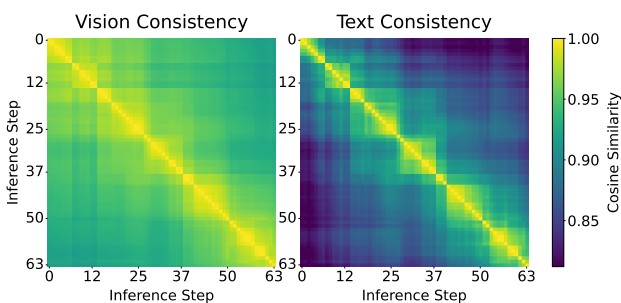

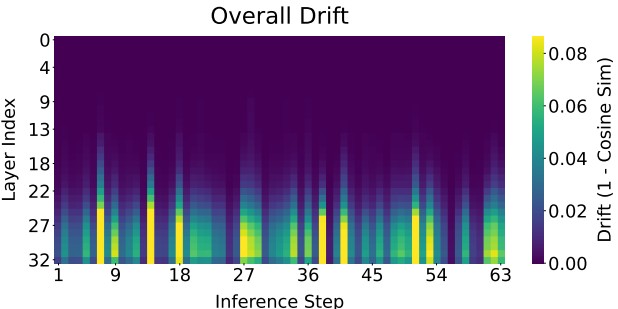

*Figure 4.* **Modality-Dependent State Evolution.** We visualize the cosine similarity matrix of hidden states across different inference steps. Left: Visual tokens exhibit high stability, indicating low drift during the decoding process. Right: Textual tokens show lower similarity between steps, indicating high volatility.

*Figure 5.* **Depth-Dependent Hidden State Drift.** The heatmap illustrates the magnitude of hidden state drift (measured as $1-$Cosine Similarity) across network layers (Y-axis) and inference steps (X-axis). Shallow layers remain stable with minimal drift, whereas deep layers exhibit significant volatility.

directly aggregate global spatiotemporal context across all frames. This mechanism enhances understanding efficiency and prevents the irreversible loss of mid-sequence event details during deep semantic abstraction, even under sparse evidence conditions. Furthermore, it inherently eliminates the visibility asymmetry that induces early receptive field in AR decoding (Proposition 3.1).

### 3.2. Training Pipeline

**Data Composition.** To address the scarcity of minute-scale, long-duration video understanding training data, we first curate a dataset centered on 2–30 minute videos. The pipeline involves (1) temporal stratification, (2) instruction synthesis via LLM, (3) text-only bias filtering, and (4) MLLM consistency voting (See Appendix E.1.1 for details).

**Multi-Stage Training.** We further adopt a multi-stage training strategy for VidLaDA (See Appendix E.1.2 for details), including (1) short-clip temporal pre-training, (2) temporal scaling warm-up, and (3) long-form video expansion.

### 3.3. Model Architecture

The overall architecture of VidLaDA, as shown in Figure 1, integrates a robust vision encoder with a DLM backbone to enable holistic spatiotemporal-friendly reasoning.

**Vision Encoder and Spatial Pooling.** We utilize SigLIP2-SO400M (Tschannen et al., 2025) to extract feature representations from the input video frames. To efficiently manage the extensive token sequence inherent to video data, we adopt a straightforward spatial pooling strategy common in recent Video LLMs (Zhang et al., 2024b; Li et al., 2024a). Specifically, after projection, the visual embeddings are reshaped to their original 2D spatial grid and downsampled via $2 \times 2$ bilinear interpolation. This operation reduces the visual sequence length by a factor of 4, balancing computational efficiency with the preservation of spatial structure.

**Bidirectional Diffusion Language Model.** The core unit of VidLaDA is the Diffusion Language Model with the full bidirectional attention mechanism. This architecture addresses the limitations of traditional Autoregressive (AR) models to some extent. Standard AR models rely on causal attention, which imposes a strict left-to-right dependency. This structure introduces the asymmetric receptive field, where early visual tokens are structurally prevented from attending to subsequent visual or textual context from user input, disrupting the global topology of the vision (See Section 3.1). In contrast, VidLaDA leverages a full bidirectional attention mechanism. This design eliminates the causal constraint, allowing every token, whether visual or textual, to attend to the holistic sequence simultaneously. This ensures that the vision encoder and language model are deeply coupled, preserving the integrity of dynamic video information.

Furthermore, unlike the serial, token-by-token generation of AR models, VidLaDA is capable of predicting multiple tokens simultaneously within a single step (Nie et al., 2025). This property significantly enhances throughput potential, making it better suited for the long-context reasoning requirements of complex spatiotemporal understanding tasks.

## 4. Efficient Video Reasoning via MARS-Cache

While DLMs offer the advantage of parallel decoding, they fundamentally differ from AR models during inference. In AR generation, past key-value pairs are cached, restricting computation to the newest token. In contrast, the bidirectional nature of vanilla DLMs necessitates re-computing attention for the entire sequence at every decoding step (Nie et al., 2025; Ye et al., 2025c; You et al., 2025). Consequently, for video understanding, the massive number of video tokens must be processed iteratively alongside the text, creating a prohibitive computational burden, thereby partially offsetting the efficiency advantages of parallel decoding. To address this, we propose the **M**ulti-modal **A**synchronous

**R**efreshing **S**trategy (**MARS**-Cache), a framework designed to prune redundancy based on the spatiotemporal behavior of DLM-based Video LLMs.

### 4.1. Empirical Observations

We analyze the internal state evolution during the decoding process and identify four distinct observations:

**(1) Chunk-wise Locality with Global Anchor Tokens.** As shown in Figure 3, inter-frame interactions primarily exhibit local dependencies. However, a specific subset of anchor tokens consistently attracts high attention scores globally. Crucially, we observe that the spatial positions of these anchor tokens remain stable across decoding steps, indicating that the global information hubs are determined early in the inference process. Furthermore, we observe a hierarchical inclusion property: the set of anchor tokens in deeper layers is a subset of those in shallow layers ($\mathcal{I}_{\text{deep}} \subset \mathcal{I}_{\text{shallow}}$), adhering to a dense-to-sparse pyramid structure.

**(2) Modality-Dependent Drift.** As shown in Figure 4, there is a notable disparity in hidden state semantic evolution between modalities. For the part of user inputs, text hidden states exhibit higher temporal drift across steps compared to vision tokens.

**(3) Depth-Dependent Stability Variance.** As shown in Figure 5, hidden state drift is not uniform across network depth. Shallow layers exhibit high stability, while deep layers show significant volatility.

**(4) Progressive Attention Sparsity.** As shown in Figure 6, the attention distribution evolves through the network depth, transitioning from a near-uniform distribution in shallow layers to a highly peaked, sparse distribution in deep layers.

### 4.2. Multi-modal Asynchronous Refreshing Strategy

**Multi-modal Asynchronous Refreshing.** Leveraging the drift disparities (Obs. 2 & 3), we introduce a hierarchical asynchronous refresh schedule. This strategy applies to context hidden states (visual tokens, prompts, decoded text).

We maintain caches for context hidden states and skip updates based on the refresh interval. We partition model layers into groups (for simplicity, we use 4 groups in this work) and assign a grouped refresh interval $\tau_{(g,m)}$: **(i) Modality-wise**: Since text drifts more (Obs. 2), we set $\tau_{(*,t)} < \tau_{(*,v)}$, refreshing visual caches less frequently. **(ii) Layer-wise**: Deep layers are more volatile (Obs. 3), so we assign smaller intervals to deeper groups (i.e., $\tau_{(g-1,*)} > \tau_{(g,*)}$).

Formally, a hidden state $\boldsymbol{H}^{t,g}$ is updated only if $t$ mod $\tau_{(g,m)} = 0$; otherwise, the cached state is reused. To preserve feature consistency across depths, we constrain the refresh interval of a shallow group to be a fixed integer multiple of the deeper group (i.e., $\tau_{(g-1,*)} = k \cdot \tau_{(g,*)}$).

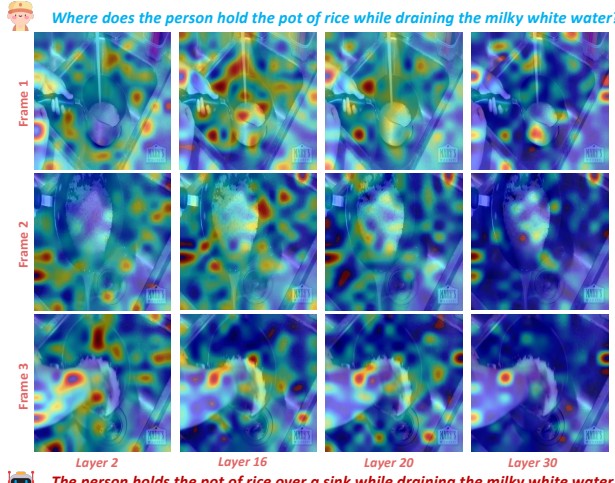

*Where does the person hold the pot of rice while draining the milky white water?*

Frame 1

Frame 2

Frame 3

Layer 2    Layer 16    Layer 20    Layer 30

*The person holds the pot of rice over a sink while draining the milky white water.*

*Figure 6.* **Visualization of Progressive Attention Sparsity.** We display the aggregated attention maps of visual tokens from the entire sequence at varying network depths. The distribution transitions from a diffuse, global pattern in shallow layers to a highly peaked, semantically focused pattern in deep layers.

This enforces a synchronization constraint: whenever a stable shallow layer is updated, the volatile deep layers are simultaneously refreshed, ensuring that low-level features are always synchronized with high-level abstractions.

**Frame-wise Chunk Attention.** While asynchronous refresh reduces the number of updates of visual tokens during inference, the update overhead of full attention for massive visual tokens still creates a major computational bottleneck when the visual cache requires refreshing. Based on Obs. 1, we observe that video tokens primarily attend to their temporal neighbors. Motivated by this sparsity, we adopt Frame-wise Chunk Attention to accelerate the inference.

Formally, we define $\mathcal{F}_n$ as the subset of visual tokens $\boldsymbol{E}^{\mathcal{V}}$ belonging to the $n$-th frame. Accordingly, the local temporal neighborhood for frame $n$ is defined as $\mathcal{N}(\mathcal{F}_n) = \mathcal{F}_{n-1} \cup \mathcal{F}_n \cup \mathcal{F}_{n+1}$. The attention for visual tokens within frame $n$ is then computed as:

$$\boldsymbol{O}_{[\mathcal{F}_n]} = \text{Softmax}\left(\frac{\boldsymbol{Q}_{[\mathcal{F}_n]}\boldsymbol{K}_{[\mathcal{N}(\mathcal{F}_n)]}^{\top}}{\sqrt{d_k}}\right)\boldsymbol{V}_{[\mathcal{N}(\mathcal{F}_n)]}, \quad (2)$$

where $d_k$ represents the dimensionality of the key, and the $\boldsymbol{O}_{[\mathcal{F}_n]}$ represents the attention outputs for frame $n$. This operation effectively reduces the total complexity of the attention mechanism between video tokens from quadratic $\mathcal{O}(|\boldsymbol{E}^{\mathcal{V}}|^2)$ to linear $\mathcal{O}(|\mathcal{N}(\mathcal{F}_n)| \times |\boldsymbol{E}^{\mathcal{V}}|)$.

Moreover, to ensure the robustness of the visual representation, we still perform full attention at the first decoding step to establish precise global context and initialize the visual cache. In subsequent steps, when the visual cache requires refreshing, we switch to the efficient chunk attention.

*Table 1.* **Comparison with SOTA models across comprehensive benchmarks.** The models are categorized into AR and DLM baselines. #S: Model Size (Parameters). #F: Number of input frames. Models marked with $*$ denote the reproduced results.

| Model | #S | #F | Video-MMMU | LongVideoBench | LVBench | EgoSchema | MVBench | MLVU$_{dev}$ | MLVU$_{test}$ | Video-MME |
|---|---|---|---|---|---|---|---|---|---|---|
| *AR Baselines* | | | | | | | | | | |
| VideoLLaMA2.1 (Cheng et al., 2024) | 7B | 32 | - | - | 36.2 | 53.1 | 57.3 | 61.2 | - | 54.9 |
| VideoChat2 (Li et al., 2025a) | 7B | 16 | - | 36.0 | - | 54.4 | - | - | - | 39.5 |
| InternVL2.5 (Chen et al., 2024d) | 7B | 64 | - | 60.0 | 38.4 | 51.5 | 72.0 | 68.9 | - | 64.2 |
| Qwen2-VL (Wang et al., 2024a) | 7B | 2fps | - | - | - | 66.7 | 67.0 | - | - | 63.3 |
| Qwen2.5-VL* (Bai et al., 2025b) | 7B | 64 | 47.4 | 60.2 | 45.3 | 65.0 | 69.6 | 62.8 | 45.3 | 63.9 |
| Video-LLaVA (Lin et al., 2024) | 7B | 8 | - | - | - | 38.4 | - | 47.3 | - | 39.9 |
| LLaVA-NeXT-Video (Zhang et al., 2024a) | 7B | 32 | - | 43.5 | - | 43.9 | 33.7 | - | - | 46.5 |
| LLaVA-OneVision* (Li et al., 2024a) | 7B | 32 | 33.9 | 56.5 | 40.7 | 60.1 | 56.7 | 64.7 | 45.3 | 58.5 |
| LLaVA-Video* (Zhang et al., 2024b) | 7B | 64 | 37.1 | 58.2 | 41.5 | 57.3 | 58.6 | 70.8 | 50.4 | 63.7 |
| *DLM Baselines* | | | | | | | | | | |
| Dream-VL (Ye et al., 2025a) | 7B | - | - | - | - | - | - | 61.1 | - | 61.5 |
| SDAR-VL (Cheng et al., 2025) | 8B | - | - | - | - | - | - | 65.0 | - | 60.8 |
| LLaDA-V* (You et al., 2025) | 8B | 32 | 43.3 | 58.6 | 36.4 | 57.9 | 53.1 | 59.4 | 44.1 | 56.4 |
| **VidLaDA (Ours)** | 8B | 64 | **46.6** | **61.4** | **44.7** | **64.5** | **59.4** | **69.2** | **53.4** | **64.1** |

**Adaptive Anchor Token Searching.** However, naively restricting global interactions through chunking compromises the long-range information flow, leading to severe performance degradation (as analyzed in Appendix F.7.1). We hypothesize that specific anchor tokens within each frame act as critical inter-frame bridges, aggregating and transmitting global visual semantics across the temporal sequence. Exclusively relying on frame-wise chunk attention impedes these pathways, disrupting the hierarchical propagation of visual information, particularly in deeper layers. To restore global connectivity efficiently, we propose an optimized searching strategy comprising the following key components:

**(i) Adaptive Proxy Scoring:** Computing the full attention map to identify anchor tokens is cost-prohibitive. Instead, we employ equidistant subsampling. Specifically, we sample a small subset of query tokens $\hat{\mathcal{S}}$ (e.g., $|\hat{\mathcal{S}}| = 32$) to compute a low-rank proxy attention matrix:

$$\hat{\boldsymbol{A}} = \text{Softmax}(\boldsymbol{Q}_{[\hat{\mathcal{S}}]}\boldsymbol{K}_{[\mathcal{S}^{\mathcal{V}}]}^{\top}/\sqrt{d_k}) \in \mathbb{R}^{|\hat{\mathcal{S}}| \times |\mathcal{S}^{\mathcal{V}}|}, \quad (3)$$

where the $\mathcal{S}^{\mathcal{V}}$ denotes the set of video tokens. To mitigate self-attention bias, we mask the entries where queries attend to themselves, yielding the debiased attention matrix $\boldsymbol{A}$.

**(ii) Temporal Reuse:** Leveraging Obs. 1, we perform this proxy scoring only at the first decoding step ($t = 1$). The identified sink indices are cached and reused for all subsequent sparse attention steps, reducing the extra computational overhead.

**(iii) Group-wise Allocation:** Leveraging the hierarchical consistency of anchor tokens in Obs. 1, we avoid per-layer searching. Instead, we search once per layer group. We allocate a decreasing budget $k_g$ for deeper groups (e.g., $k_{g-1} > k_g > \dots$). For group $g$, we aggregate the importance scores from the proxy matrix $\boldsymbol{A}^{g_{\text{start}}}$ (computed at the group's first layer) and select the top-$k_g$ indices.

To ensure a contiguous memory layout and consistent computational load, we enforce a fixed budget of anchor tokens per frame. For each frame $\mathcal{F}_j$, we aggregate the importance scores across the sampled queries. Consequently, at step $t = 1$, the anchor indices $\mathcal{I}_{(g,\mathcal{F}_j)}$ for each frame within group $g$ are determined by:

$$\boldsymbol{A}' = \sum_i \boldsymbol{A}^{g_{\text{start}}}_{[i,\mathcal{F}_j]} \in \mathbb{R}^{|\mathcal{F}_j|}, \; \mathcal{I}_{(g,\mathcal{F}_j)} = \text{Top}_{k_g}\left(\boldsymbol{A}'\right). \quad (4)$$

The final set of global anchor tokens, $\mathcal{I}_g = \bigcup_j \mathcal{I}_{(g,\mathcal{F}_j)}$, is subsequently made visible to all tokens in the sequence, while these anchor tokens retain the ability to attend to all other tokens, effectively restoring global information propagation. Finally, exploiting the permutation invariance of bidirectional attention, we can relocate the anchor tokens within each frame to the beginning of the sequence to optimize memory access patterns.

# 5. Experiments

In this section, we first evaluate VidLaDA against state-of-the-art (SOTA) AR and DLM baselines across diverse benchmarks. Then, we investigate the performance of Chain-of-Thought (CoT) inference combined with MARS-Cache. Finally, this section concludes with a series of ablation studies. Detailed experimental setups are provided in Appendix G.1.

## 5.1. Main Results

We benchmark VidLaDA against the current leading open-source SOTA AR and DLM baselines. The comparative results are presented in Table 1.

As the first DLM-based Video LLM optimized with bidirectional spatiotemporal attention, VidLaDA establishes a new baseline for non-autoregressive video understanding. It consistently outperforms existing DLM approaches across all

*Table 2.* **Evaluation of MARS-Cache Acceleration on Video Reasoning.** We assess the generalizability and efficiency of MARS-Cache by applying it under Chain-of-Thought (CoT) settings. **TPS** denotes Throughput (Tokens Per Second).

| Model | Ego$_{subset}$ | | MLVU$_{test}$ | | LongVideoBench | |
|---|---|---|---|---|---|---|
| | Acc | TPS | Acc | TPS | Acc | TPS |
| LLaVA-OV (CoT) | 62.8 | 27.0 | 43.3 | 27.2 | 57.5 | 25.0 |
| LLaDA-V (CoT) | 65.4 | 3.5 | 43.2 | 4.3 | 58.1 | 4.5 |
| + Dual-Cache | 64.4 | 23.8 | 43.2 | 28.3 | 59.2 | 27.2 |
| **+ MARS-Cache** | **65.2** | **35.2** | **43.6** | **40.0** | **59.6** | **37.9** |
| VidLaDA (CoT) | 67.4 | 2.7 | 50.2 | 3.3 | 59.8 | 2.1 |
| + Dual-Cache | **67.2** | 23.7 | 50.5 | 26.0 | 59.4 | 18.5 |
| **+ MARS-Cache** | 67.0 | **33.6** | **50.7** | **33.6** | **59.7** | **25.2** |

evaluated dimensions (LLaDA-V (You et al., 2025), SDAR-VL (Cheng et al., 2025), Dream-VL (Ye et al., 2025a)).

Furthermore, VidLaDA demonstrates remarkable competitiveness against top-tier open-sourced AR-based Video LLMs, including Qwen2.5-VL (Bai et al., 2025b), LLaVA-OneVision (Li et al., 2024a), and LLaVA-Video (Zhang et al., 2024b). Notably, VidLaDA exhibits strengths particularly in tasks necessitating complex spatiotemporal understanding, where unidirectional modeling may constrain the aggregation of distributed visual evidence. For instance, it surpasses LLaVA-Video and LLaVA-OneVision on LongVideoBench and outperforms the robust Qwen2.5-VL on MLVU (both Dev and Test splits). These results support the hypothesis that our bidirectional attention mechanism effectively addresses the asymmetric receptive field limitations of AR architectures by enabling more robust global dependency modeling (See Section 3.1). To separate this broad SOTA comparison from frame-count effects, Appendix F.3 reports VidLaDA results from 8 to 64 frames; at the matched 32-frame setting, VidLaDA still outperforms LLaVA-OneVision across all reported Table 1 benchmarks.

## 5.2. Results under CoT Inference

To unlock the complex reasoning capabilities of models, we employ a structured Chain-of-Thought (CoT) inference pipeline inspired by (Zhou et al., 2024). This pipeline proceeds through four distinct stages: (1) task prompt routing, (2) reasoning analysis, (3) self-reflection, and (4) final answer generation. Further details on the specific prompt designs and experimental setup are provided in Appendix E.2 and Appendix G.4, respectively.

We evaluate the efficiency and robustness of our proposed strategies on EgoSchema (subset), MLVU (test), and LongVideoBench. Quantitative results are summarized in Table 2, with the batch size set to 1. To ensure practical inference speeds and establish a strong baseline, we employ parallel decoding (Wu et al., 2025) by default across all

DLM settings. The primary objective of this experiment is to verify the generalizability of MARS-Cache on DLMs (LLaDA-V and VidLaDA) under CoT-style inference, and to benchmark its efficiency against the AR baseline.

As shown in Table 2, MARS-Cache delivers consistent and substantial throughput gains across DLMs and all benchmarks, yielding ∼8-12× speedups over vanilla DLM decoding. Importantly, with MARS-Cache, DLMs can even surpass the throughput level of AR (∼1.3× TPS). Moreover, MARS-Cache consistently outperforms Dual-Cache (Wu et al., 2025), providing an additional ∼1.3-1.5× TPS improvement by pruning redundant computations via modal-wise cache refreshing and frame-wise chunk attention. Crucially, despite the aggressive reuse of intermediate states, MARS-Cache largely preserves reasoning accuracy. Notably, it achieves performance improvements in several settings, while maintaining comparable results on other benchmarks with only negligible fluctuations.

Overall, these results demonstrate that MARS-Cache substantially accelerates CoT-style video reasoning and generalizes well across DLM-based Video LLMs. Some visual examples of the reasoning trajectories generated with MARS-Cache are detailed in Appendix F.5.

## 5.3. Ablation Studies

In this section, we summarize ablations that examine the architectural effect of bidirectional DLMs under matched fine-tuning and analyze the key design components of MARS-Cache.

### 5.3.1. CONTROLLED ARCHITECTURE COMPARISON

To reduce training confounders, we compare AR and DLM baselines under matched fine-tuning settings. Table 3 shows that the same fine-tuning recipe lowers the AR average while preserving the DLM gain after matched fine-tuning. Under matched conditions, this controlled response suggests that the DLM architecture may utilize the available spatiotemporal evidence more effectively than the AR counterpart, which is consistent with our understanding efficiency analysis in Proposition 3.2. Detailed settings and further diagnostics are provided in Appendix F.6.

This controlled comparison further contextualizes the broad SOTA results in Table 1. Following standard MLLM evaluation practice, Table 1 compares complete systems that may differ in backbones and training recipes; by contrast, the matched fine-tuning control reduces these confounding factors to some extent and provides additional evidence for the role of the bidirectional DLM architecture.

### 5.3.2. ABLATING MARS-CACHE COMPONENTS

We next summarize the main findings for the three key

*Table 3.* **Results of Architectural Comparisons.** Effect of extending training for AR vs. DLM baselines under the same training dataset and identical hyperparameters.

| Model | #F | LongVideoBench | LVBench | MVBench | Video-MME | Avg |
|---|---|---|---|---|---|---|
| *AR Baselines* | | | | | | |
| LLaVA-OneVision | 32 | 56.5 | 40.7 | 56.7 | 58.5 | 53.1 |
| +SFT 1 Epoch | 32 | 57.4 (+0.9) | 38.8 (-1.9) | 54.1 (-2.6) | 58.6 (+0.1) | 52.2 (-0.9) |
| +SFT 4 Epoch | 32 | 56.0 (-0.5) | 38.3 (-2.4) | 54.5 (-2.2) | 56.8 (-1.7) | 51.4 (-1.7) |
| *DLM Baselines* | | | | | | |
| LLaDA-V | 32 | 58.2 | 36.4 | 53.1 | 56.4 | 51.0 |
| +SFT 1 Epoch | 32 | 58.9 (+0.7) | 36.6 (+0.2) | 53.7 (+0.6) | 58.5 (+2.1) | 51.9 (+0.9) |
| +SFT 4 Epoch | 32 | 58.0 (-0.2) | 37.1 (+0.7) | 53.6 (+0.5) | 58.7 (+2.3) | 51.9 (+0.9) |

components of MARS-Cache, with comprehensive results provided in Appendix F.7.

**(i) The Necessity of Anchor Token:** Our assessment of global connectivity reveals that retaining anchor tokens is crucial; completely reverting to frame-wise chunk attention significantly degrades performance. We find that preserving full attention in shallow layers to maintain structural integrity while aggressively pruning anchors in deep layers achieves the optimal trade-off between reasoning accuracy and computational cost.

**(ii) Asynchronous Refreshing Strategy:** Leveraging the observation that visual hidden states exhibit higher temporal stability than textual ones, we implement a modality-wise refreshing strategy. We conclude that setting the visual refresh interval larger than the textual interval (optimally $R_{v/t} \approx 2$) significantly boosts throughput without compromising accuracy, whereas excessive ratios lead to degradation. Additionally, regarding network depth, a pyramid-style schedule (increasing update frequency for deeper, more volatile layers) consistently outperforms uniform schedules and is orthogonal to the modality-wise refreshing strategy.

**(iii) Search Overhead:** For the adaptive anchor searching mechanism, we determine that a query subset size of 128 offers a robust compromise. It is sufficient to identify high-quality global information hubs while keeping the computational overhead negligible, whereas using the full sequence leads to memory bottlenecks.

## 6. Conclusion

In this work, we introduce VidLaDA, a bidirectional diffusion language model framework that overcomes the spatiotemporal limitations of AR-based Video LLMs. By enabling global bidirectional attention, VidLaDA overcomes the inherent limitations of causal decoding, allowing more efficient and balanced aggregation of spatiotemporal visual evidence for robust video understanding. To mitigate the

computational overhead of DLM-based Video LLM decoding, we further introduce MARS-Cache, an asynchronous cache refreshing strategy that exploits the stability of visual features and frame-wise locality to substantially accelerate inference. Extensive experiments demonstrate that VidLaDA outperforms existing DLM baselines and competes with SOTA AR-based models, establishing a new, efficient paradigm for non-autoregressive video understanding.

## Acknowledgements

The paper is supported in part by the National Natural Science Foundation of China (No.62325109, 62595733, 62561160155), and in part by the Shanghai 'The Belt and Road' Young Scholar Exchange Grant (24510742000).

## Impact Statement

This work improves the efficiency of video understanding, which may also lower the cost of processing sensitive video data at scale. Deployments should use authorized data, respect privacy and consent, and avoid applications such as unauthorized tracking, profiling, or surveillance-like monitoring of individuals. Our goal is to advance research on video understanding and efficient inference, not to encourage unconstrained deployment in privacy-sensitive settings.

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

# A. Main Notation Introduction

*Table 4.* Meanings of the Main Notations

| Notation | Size | Meaning |
|---|---|---|
| $\lvert * \rvert$ | Constant | The length of the corresponding sequence. |
| $d$ | Constant | The model dimension. |
| $d_k$ | Constant | The dimensionality of the key in the multi-head mechanism. |
| $\mathcal{V}$ | $F \times C \times H \times W$ | The input video frames. |
| $\boldsymbol{E}^{\mathcal{V}}$ | $\lvert \boldsymbol{E}^{\mathcal{V}} \rvert \times d$ | The spatiotemporal visual tokens encoded by the ViT. |
| $P$ | $\lvert P \rvert$ | The token IDs of the text prompt produced by the tokenizer. |
| $R$ | $\lvert R \rvert$ | The token IDs of the text response produced by the tokenizer. |
| $\boldsymbol{z}_t$ | $d$ | The latent representation encoded by ViT and LLM at timestep $t$. |
| $\mathcal{I}$ | $\lvert \mathcal{I} \rvert$ | The index set of anchor tokens. |
| $\tau_{(g,m)}$ | Constant | The refresh interval in group $g$ for modality $m$. |
| $\boldsymbol{H}$ | $\lvert \boldsymbol{H} \rvert \times d$ | The hidden states in the LLMs. |
| $\mathcal{F}_n$ | $\lvert \mathcal{F}_n \rvert$ | The index set of visual tokens $\boldsymbol{E}^{\mathcal{V}}$ belonging to the $n$-th frame. |
| $\boldsymbol{O}$ | $\lvert \boldsymbol{O} \rvert \times d_k$ | The attention output matrix. |
| $\boldsymbol{Q}$ | $\lvert \boldsymbol{Q} \rvert \times d_k$ | The attention query matrix. |
| $\boldsymbol{K}$ | $\lvert \boldsymbol{K} \rvert \times d_k$ | The attention key matrix. |
| $\boldsymbol{V}$ | $\lvert \boldsymbol{V} \rvert \times d_k$ | The attention value matrix. |
| $\hat{\mathcal{S}}$ | $\lvert \hat{\mathcal{S}} \rvert$ | The index set of query tokens obtained by anchor token searching. |
| $\mathcal{S}^{\mathcal{V}}$ | $\lvert \mathcal{S}^{\mathcal{V}} \rvert$ | The index set of video tokens. |

# B. Detailed Problem Formulation and Background

In this section, we provide the mathematical formulation for the Autoregressive baseline and the Diffusion Language Model backbone used in VidLaDA.

## B.1. Autoregressive-Based Video Language Models

The prevailing paradigm for Multimodal Large Language Models (MLLMs) treats video understanding as a conditional sequence generation task. Given visual tokens $\boldsymbol{E}^{\mathcal{V}}$ (derived from video $\mathcal{V}$ via a ViT and projector) and prompt tokens $P$, the model maximizes the conditional likelihood of the response $R$ autoregressively:

$$\mathcal{L}_{\text{AR}} = -\sum_{i=1}^{\lvert R \rvert} \log p_\theta(R_i \mid \boldsymbol{E}^{\mathcal{V}}, P, R_{<i}), \tag{5}$$

where $p_\theta$ is parameterized by a Transformer Decoder (Vaswani et al., 2017; Grattafiori et al., 2024). This approach necessitates a causal attention mask $\boldsymbol{M}$, defined as:

$$\boldsymbol{M}_{ij} = \begin{cases} 0 & \text{if } j \le i, \\ -\infty & \text{otherwise.} \end{cases} \tag{6}$$

This factorization imposes a strictly unidirectional dependency, preventing visual or early text tokens from attending to future tokens.

## B.2. Diffusion Language Models (DLMs)

DLMs (Nie et al., 2025; Ye et al., 2025c) formulate generation as a discrete diffusion process.

**Forward Process.** In the context of conditional generation, the forward process adds noise to the response $R$ while keeping the conditions (visual inputs $\mathcal{V}$ and text prompt $P$) unmasked. Let $R^0$ denote the clean response sequence. At continuous time $t \in [0, 1]$, each token $R_i^t$ in the response is independently replaced by [MASK] with probability $t$. The transition distribution is defined specifically over the response tokens:

$$q^{t|0}(R^t \mid R^0) = \prod_{i=1}^{|R|} q^{t|0}(R_i^t \mid R_i^0), \tag{7}$$

$$\text{where } q^{t|0}(R_i^t \mid R_i^0) = \begin{cases} 1 - t & \text{if } R_i^t = R_i^0, \\ t & \text{if } R_i^t = \text{[MASK]}. \end{cases} \tag{8}$$

**Reverse Process and Objective.** The reverse process aims to recover the clean response $R^0$ from the masked state $R^t$. A neural network $p_\theta(\cdot \mid \boldsymbol{E}^{\mathcal{V}}, P, R^t)$ predicts the original tokens for masked positions within the response, conditioned on the fully observable $\boldsymbol{E}^{\mathcal{V}}$ and $P$. The objective minimizes the variational upper bound:

$$\mathcal{L}_{\text{DLM}} = -\mathbb{E}_{t,R^0,R^t} \left[ \frac{1}{t} \sum_{i=1}^{|R|} m_i^t \log p_\theta(R_i^0 \mid \boldsymbol{E}^{\mathcal{V}}, P, R^t) \right], \tag{9}$$

where $m_i^t = \mathbb{I}[R_i^t = \text{[MASK]}]$ is the mask indicator.

**Bidirectional Attention.** Unlike AR models, the prediction of $R_i^0$ and understanding of $\boldsymbol{E}^{\mathcal{V}}, P, R$ in DLMs depend on the global context. Thus, we utilize full bidirectional attention:

$$\text{Attention}(\boldsymbol{Q}, \boldsymbol{K}, \boldsymbol{V}) = \text{Softmax}\left(\frac{\boldsymbol{Q}\boldsymbol{K}^{\top}}{\sqrt{d_k}}\right)\boldsymbol{V}. \tag{10}$$

This mechanism enables global spatiotemporal modeling for $\boldsymbol{E}^{\mathcal{V}}$. During inference, the model starts from a fully masked $R^1$ and iteratively refines the sequence by predicting $R^0$ and re-masking based on a schedule.

## C. Related Work

### C.1. Autoregressive-Based Video Large Language Models

Following the success of Large Language Models (LLMs) in text generation, Multimodal Large Language Models (MLLMs) have rapidly evolved to perceive visual inputs. Early works like LLaVA (Liu et al., 2024b) projected static visual features into the LLM's token space, enabling image-text question answering. This paradigm was quickly adapted to the video domain, deriving the Video Large Language Models (Video LLMs). Models such as Video-LLaVA (Lin et al., 2024), VideoChat (Li et al., 2025a), VideoLLaMA (Cheng et al., 2024), LongVILA (Chen et al., 2024c), InternVideo (Wang et al., 2024c), mPLUG-Owl (Ye et al., 2024), VideoLoom (Shi et al., 2026), and PLLaVA (Xu et al., 2024) aggregate temporal frames using pooling or Q-Former (Li et al., 2023a) structures before feeding them into an autoregressive LLM decoder. More recent state-of-the-art models, including LLaVA-OneVision (Li et al., 2024a), LLaVA-Video (Zhang et al., 2024b), ProLongVid (Wang et al., 2025b), Qwen2-VL (Wang et al., 2024a), and Qwen2.5-VL (Bai et al., 2025b) scale up the visual resolution and context length, achieving impressive performance on general video benchmarks.

However, these models fundamentally rely on the Autoregressive (AR) Model, based on the causal masking mechanism. As analyzed in Section 3.1, causal masking imposes a unidirectional dependency where visual tokens cannot attend to subsequent context during the encoding of the history. This structural asymmetry limits the modeling of global spatiotemporal dependencies and often leads to context fading in video understanding. VidLaDA addresses this by replacing the AR backbone with a bidirectional diffusion model, ensuring omnidirectional information flow.

We further note that the anchor tokens in MARS-Cache should not be conflated with high-attention artifacts in causal AR models. StreamingLLM (Xiao et al., 2024) identifies attention sinks induced by the universal visibility of early prefix tokens under causal masking. DyToK (Li et al., 2026) observes temporal attention outliers in AR MLLMs and uses attention signals to guide dynamic token compression. In contrast, VidLaDA uses bidirectional attention, and MARS-Cache explicitly preserves anchor tokens as visual information hubs to maintain global connectivity under frame-wise sparse attention.

## C.2. Diffusion and Non-Autoregressive Language Models

Diffusion models have achieved remarkable success in continuous data generation (Peebles & Xie, 2023; Polyak et al., 2024; Li et al., 2023b; Wang et al., 2024b; Gan et al., 2025). Recently, substantial progress has been made in adapting diffusion to discrete language modeling, including diffusion in continuous embedding space (Li et al., 2022) and masked/discrete formulations such as D3PM (Ho et al., 2020) and SSD-LM (Han et al., 2023). More recent large-scale Diffusion Language Models (DLMs), e.g., LLaDA (Nie et al., 2025), LLaDA-MoE (Zhu et al., 2025), LLaDA2 (Bie et al., 2025), and Dream (Ye et al., 2025c), demonstrate competitive generation quality to strong autoregressive LLMs while enabling parallel decoding via iterative denoising. In multimodal settings, several works extend DLMs to vision-language tasks by conditioning on spatiotemporal tokens and largely retaining the standard MLLM pipeline (vision encoder → projector → language backbone), where diffusion is introduced mainly by swapping the AR backbone with a masked-denoising Transformer (e.g., LLaDA-V (You et al., 2025), Dream-VL (Ye et al., 2025a), and concurrent efforts (Li et al., 2025b; Cheng et al., 2025; Xin et al., 2025)).

While these studies validate the feasibility of DLM-based MLLMs and acknowledge the benefits of bidirectional context, our work differentiates itself by offering a formalized theoretical analysis (Propositions 3.1 and 3.2) and rigorous empirical analysis. We explicitly target the spatiotemporal challenges unique to video understanding to demonstrate *"Why DLM is structurally more suitable than AR in image/video understanding?"*, beyond general generative capabilities. VidLaDA bridges diffusion modeling and video understanding in a more video-oriented manner. We first analyze and justify why bidirectional DLM is suitable for spatiotemporal understanding in videos, beyond treating diffusion as a drop-in replacement for AR decoding. Building on this motivation, we present (to the best of our knowledge) the first DLM-based Video LLM that demonstrates strong performance on complex video understanding benchmarks. Finally, to make diffusion inference practical under long-form video contexts with massive video tokens, we introduce MARS-Cache, a multimodal, structure-aware caching framework that significantly reduces redundant computation during decoding.

# D. Proof

## D.1. Proof of Proposition 3.1

Consider a sequence $\boldsymbol{X} = \{\boldsymbol{x}_1, \ldots, \boldsymbol{x}_{|\boldsymbol{X}|}\}$ processed by a causal attention mechanism. The attention score $\boldsymbol{A}_{i,j}$ (the attention $\boldsymbol{x}_i$ pays to $\boldsymbol{x}_j$) is valid only if $j \leq i$. This constraint is enforced by a lower-triangular causal mask $\boldsymbol{M} \in \{0, -\infty\}^{T \times T}$.

We define the visibility frequency $\mathcal{C}(\boldsymbol{x}_j)$ of a token $\boldsymbol{x}_j$ as the number of times it acts as a key/value for other tokens in the sequence during a forward pass:

$$\mathcal{C}(\boldsymbol{x}_j) = \sum_{t=1}^{|\boldsymbol{X}|} \mathbb{I}(j \leq t) = |\boldsymbol{X}| - j + 1, \tag{11}$$

where $\mathbb{I}$ is the indicator function.

For the initial token $\boldsymbol{x}_1$, $\mathcal{C}(\boldsymbol{x}_1) = |\boldsymbol{X}|$, meaning it is visible to and attended by every token in the sequence. In contrast, for a later token $\boldsymbol{x}_k$ (where $k \gg 1$), $\mathcal{C}(\boldsymbol{x}_k)$ is significantly smaller.

Because the softmax function $\sigma(\boldsymbol{x}_j) = e^{\boldsymbol{x}_j} / \sum_k e^{\boldsymbol{x}_k}$ forces weights to sum to 1, the model requires a stable token to absorb excess attention mass when current semantic features are ambiguous (Xiao et al., 2024). Since $\boldsymbol{x}_1$ is the only universally accessible token with maximum visibility frequency, the model learns an inductive bias to utilize $\boldsymbol{x}_1$ as a specialized asymmetric receptive field.

For the visual sequence $\boldsymbol{E}^{\mathcal{V}}$, since the index $j = 1$ corresponds to a specific spatial location (e.g., top-left corner) due to rasterization, resulting in $\mathcal{C}(\boldsymbol{e}_1^{\mathcal{V}}) = |\boldsymbol{E}^{\mathcal{V}}|$, the learned bias induces a spatially non-uniform importance distribution: the visual start token is disproportionately weighted solely due to its position in the input sequence, thereby shaping the early receptive field.

## D.2. Proof of Proposition 3.2

This section establishes the formal proof regarding the information capacity gap between the Autoregressive (AR) and Bidirectional (Bidi) decoding paradigms.

Let $\mathcal{V}$ be the random variable representing the input video. We denote the encoded visual feature sequence produced by the Vision Transformer (ViT) as $\boldsymbol{E}^{\mathcal{V}} = \{e_1^{\mathcal{V}}, \ldots, e_{|\boldsymbol{E}^{\mathcal{V}}|}^{\mathcal{V}}\}$, assuming it preserves the global information of $\mathcal{V}$. We model the decoding process as a Markov chain where $z_t$ is the latent state at the final layer of the LLM at position $t$. The Bidirectional paradigm utilizes full self-attention, characterized by the Markov chain $\mathcal{V} \to \boldsymbol{E}^{\mathcal{V}} \to z_t^{\text{bidi}}$. Conversely, the Autoregressive paradigm (AR) is conditioned strictly on the history $\boldsymbol{E}^{\mathcal{V}}_{\leq t} = \{e_1^{\mathcal{V}}, \ldots, e_t^{\mathcal{V}}\}$, characterized by $\mathcal{V} \to \boldsymbol{E}^{\mathcal{V}}_{\leq t} \to z_t^{\text{AR}}$.

To quantify the disparity, we define the *cumulative pointwise information capacity* as $\mathcal{K} \triangleq \sum_{t=1}^{T} I(z_t; \mathcal{V})$. This metric represents the aggregate source information accessible across the generated sequence. Our analysis rests on the following two theoretical assumptions.

**Assumption D.1.** Visual information is spatially distributed and the encoder is non-redundant. Specifically, for any $t < T$, the future visual features $\boldsymbol{E}^{\mathcal{V}}_{>t}$ contain non-zero conditional mutual information about $\mathcal{V}$:

$$I(\boldsymbol{E}^{\mathcal{V}}_{>t}; \mathcal{V} \mid \boldsymbol{E}^{\mathcal{V}}_{\leq t}) > 0.$$

**Assumption D.2.** The LLM decoder possesses sufficient expressivity such that the information bottleneck is dominated by architectural visibility constraints. Consequently, the supremum of the mutual information is bounded by the Data Processing Inequality (DPI):

$$\sup I(z_t^{\text{bidi}}; \mathcal{V}) = I(\boldsymbol{E}^{\mathcal{V}}; \mathcal{V}), \tag{12}$$

$$\sup I(z_t^{\text{AR}}; \mathcal{V}) = I(\boldsymbol{E}^{\mathcal{V}}_{\leq t}; \mathcal{V}). \tag{13}$$

Proceeding with the derivation, we compare the summation terms for the two paradigms. Applying the DPI to the Markov chains defined above and invoking Assumption D.2, the information capacity gap at step $t$ is given by:

$$\Delta_t \triangleq \sup I(z_t^{\text{bidi}}; \mathcal{V}) - \sup I(z_t^{\text{AR}}; \mathcal{V}) = I(\boldsymbol{E}^{\mathcal{V}}; \mathcal{V}) - I(\boldsymbol{E}^{\mathcal{V}}_{\leq t}; \mathcal{V}). \tag{14}$$

Using the chain rule for Mutual Information, we decompose the joint information of sequence $\boldsymbol{E}^{\mathcal{V}}$ as:

$$I(\boldsymbol{E}^{\mathcal{V}}; \mathcal{V}) = I(\boldsymbol{E}^{\mathcal{V}}_{\leq t}; \mathcal{V}) + I(\boldsymbol{E}^{\mathcal{V}}_{>t}; \mathcal{V} \mid \boldsymbol{E}^{\mathcal{V}}_{\leq t}). \tag{15}$$

Substituting this yields:

$$\Delta_t = I(\boldsymbol{E}^{\mathcal{V}}_{>t}; \mathcal{V} \mid \boldsymbol{E}^{\mathcal{V}}_{\leq t}). \tag{16}$$

Under Assumption D.1, it holds that $\Delta_t > 0$ for all $t < T$. Finally, summing over the sequence length $T$ reveals a strict gap in cumulative capacity:

$$\sum_{t=1}^{T} \sup I(z_t^{\text{bidi}}; \mathcal{V}) - \sum_{t=1}^{T} \sup I(z_t^{\text{AR}}; \mathcal{V}) = \sum_{t=1}^{T-1} \Delta_t + \Delta_T$$

$$= \sum_{t=1}^{T-1} I(\boldsymbol{E}^{\mathcal{V}}_{>t}; \mathcal{V} \mid \boldsymbol{E}^{\mathcal{V}}_{\leq t}) + 0$$

$$> 0.$$

# E. Method Details

## E.1. Training Pipeline

### E.1.1. DATA COMPOSITION

To remedy the lack of minute-scale, long-horizon temporal understanding in existing video instruction-tuning data, we construct a corresponding dataset through a robust construction pipeline. Utilizing FineVideo (Farré et al., 2024) as the seed source, we curate a high-quality dataset specifically emphasizing videos ranging from 2 to 30 minutes. Our pipeline leverages Deepseek-V3.1 (Liu et al., 2024a) as the LLM for instruction synthesis and text-based filtering, and Qwen3-VL-235B-A22B (Bai et al., 2025a) as the MLLM for visual consistency verification. The pipeline consists of four distinct stages:

**Data Acquisition and Temporal Stratification.** Initial curation involves rigorous filtering for corruption and availability. To ensure balanced temporal coverage, we stratify the videos into five duration buckets (0-30s, 30s-60s, 1m-2m, 2-10m, 10-30m), with a strategic focus on processing the underrepresented 2-30 minute long-form intervals.

**Automated Instruction Synthesis.** Leveraging the rich metadata associated with the videos from FineVideo, we employ Deepseek-V3.1 to synthesize a diverse set of instruction-following tasks with the prompts from LLaVA-Video (Zhang et al., 2024b). These include Multiple Choice Questions (MCQ) and Open-ended QA, aiming to capture both fine-grained visual details and high-level narrative comprehension.

**De-biasing via Text-Only Filtering.** To mitigate text-only bias, where questions can be answered solely via subtitles or audio transcripts (e.g., news anchors, podcasts), we implement a filtering mechanism that only uses LLM to process data. We feed the generated questions into the LLM without visual inputs. Samples where the LLM correctly predicts the answer are discarded. This step effectively removes low-visual-information samples and ensures that the dataset strictly requires visual perception for reasoning.

**Quality Assurance via Consistency Voting.** To guarantee label reliability and minimize hallucinations, we employ a self-consistency voting strategy. The MLLM generates responses for each sample three times under varying temperatures. An LLM evaluator then compares these responses against the reference answers derived from the instruction synthesis phase. Only samples where the MLLM achieves a consistency consensus (i.e., at least 2 out of 3 responses match the ground truth) are retained for the final training set.

Beyond the newly curated video data, we integrate established high-quality benchmarks to ensure robust performance across diverse modalities. Specifically, we adopt LLaVA-Video-178K (Zhang et al., 2024b) as the foundational source for short-to-medium video instruction tuning, ensuring coverage of rich temporal dynamics and general video understanding. Furthermore, to maintain and enhance fine-grained spatial reasoning capabilities, we incorporate the MAmmoTH-VL (Guo et al., 2025a) dataset for image-based instruction following. This holistic data composition, which spans static images, short clips, and our proposed parts, forms the basis of our unified training curriculum.

### E.1.2. MULTI-STAGE TRAINING STRATEGY

*Table 5.* **Detailed Specifications of the Multi-Stage Training Curriculum.** We outline the data composition, temporal resolution scaling, and optimization configurations across three stages, transitioning from short-clip alignment to long-form video understanding.

|  | Stage 1 | Stage 2 | Stage 3 |
|---|---|---|---|
| #Frames | 32 | 64 | 64 |
| Max Sequence Length | 8K | 16K | 16K |
| Video Sources | Finevideo (Farré et al., 2024), ActivityNet-QA (Yu et al., 2019), NextQA (Xiao et al., 2021), Youtube (Zhu et al., 2023), ShareG-PTVideo (Zhang et al., 2025), Ego4D (Grauman et al., 2022), Perception Test (Patraucean et al., 2023), Charades (Sigurdsson et al., 2016), YouCook2 (Zhou et al., 2018) | | |
| #Sample | 1.8M | 500K | 500K |
| Min Duration | ∼10s | ∼1min | ∼1min |
| Max Duration | ∼3min | ∼3min | >30min |
| Image Ratio | 10% | 10% | 10% |
| Text Ratio | 0% | 0% | 10% |
| Trainable Modules | ViT/MLP/LLM | ViT/MLP/LLM | MLP/LLM |
| Learning Rate | 2e–6/1e–5/1e–5 | 2e–6/1e–5/1e–5 | 2e–6/2e–6 |

We employ a three-stage curriculum learning strategy with the initial checkpoint from LLaDA-V (You et al., 2025), focusing on effectively adapting the model from static image to long-form video understanding. This approach progressively scales the temporal resolution and context length, ensuring stable convergence while mitigating the catastrophic forgetting of short-term temporal dynamics. The overall recipe is listed in the Table 5.

**Stage 1: Short-Clip Temporal Pre-Training.** The primary objective of the initial stage is to equip the static MLLM with fundamental temporal perception capabilities. We utilize a large-scale collection of 1.8M short videos and images, with durations spanning from approximately 10 seconds to 3 minutes, sourced from diverse datasets. In this phase, the model is trained with a temporal resolution of 32 frames and a maximum sequence length of 8K tokens. We perform full-parameter fine-tuning on the Vision Transformer (ViT), Projector (MLP), and LLM backbone to align the vision encoder's temporal features with the language space using learning rates of 2e-6, 1e-5, and 1e-5, respectively.

**Stage 2: Temporal Scaling Warm-up.** To bridge the gap between the short and long durations of the same video, we utilize an intermediate warm-up stage that transitions the model to higher temporal resolutions. We select 500K samples from the same dataset in Stage 1, and double the temporal sampling density to 64 frames while expanding the context window to 16K tokens. We continue with full-parameter tuning using the same learning rates as Stage 1, but shift the focus to stabilizing the adaptation to longer sequence dependencies. This stage is used to prevent performance degradation on short-term actions while preparing the attention mechanism for extended temporal spans.

**Stage 3: Long-Form Video Expansion.** The final stage focuses on reasoning over long and ultra-long videos. We construct a dataset of 500K samples by combining our high-quality long-form dataset (durations of 2-30+ minutes) with some samples from Stage 2. To ensure stability, we freeze the ViT and exclusively fine-tune the MLP projector and LLM backbone with a reduced learning rate of 2e-6. Additionally, we incorporate a mixture of 10% text-only instructions into the training dataset to preserve the general instruction-following capabilities of Video LLM.

### E.2. Chain of Thought Prompts Details

To unlock the complex reasoning capabilities of Video LLMs, we employ a structured, multi-stage inference pipeline designed to emulate human cognitive processes inspired by (Zhou et al., 2024). This pipeline decomposes video understanding into four distinct phases: (1) **Task Prompt Routing**, where the question is classified into a specific reasoning domain; (2) **Reasoning Analysis**, where a specialized prompt generates intermediate reasoning steps (e.g., visual scans or timelines) without immediately answering the question; (3) **Self-Reflection**, where the model evaluates the validity of its own analysis; and (4) **Final Answer Generation**, where the verified analysis is synthesized into a natural response.

The specific prompt templates used for each stage are detailed in Tables 14 through 17.

## F. More Experimental Results

### F.1. The Role of Video Dataset Duration

This section complements Appendix E.1.1 by analyzing how the duration distribution of training videos affects video understanding. We follow the training recipe in Table 13, adopt the learning rate and trainable modules from Stage 3 of Table 5, and use the Stage 2 checkpoint as the baseline. We then construct two subsets by duration: 1min-3min and 2min-30min. For each range, we uniformly sample 20K samples and train with the same hyperparameters and input configuration, so the only variable is the duration distribution.

As shown in Table 6, the 1min-3min subset consistently underperforms the baseline, particularly on LongVideoBench (62.0 to 59.8). This indicates that restricting training to short clips limits the temporal receptive field, causing the model to overfit to short-term dependencies and degrade on long-range reasoning tasks. Conversely, the 2min-30min subset excels in temporal modeling, surpassing the baseline on Video-MME (62.1 to 64.1), LVBench, and MVBench. The improvement on Video-MME confirms that exposure to extended temporal spans is essential for learning to bridge distant visual cues, a capability not achievable through short clips alone.

However, the 2min-30min subset slightly trails the baseline on LongVideoBench (62.0 to 61.6). This may suggest a trade-off: long-duration data enhances temporal depth, while short-duration data contributes to local semantic perception. Thus, prioritizing long-duration data while retaining a mixed distribution offers the optimal balance for broad generalization.

### F.2. Multi-Stage Training

This section empirically validates the efficacy of the multi-stage curriculum detailed in Section E.1.2 by evaluating checkpoints at the end of each training stage. Table 7 presents the quantitative progression of the model performance and the corresponding GPU-hour cost.

*Table 6.* **Comparison with Different Video Dataset Duration.**

| Model | #F | LongVideoBench | LVBench | MVBench | Video-MME |
|-------|----|----------------|---------|---------|-----------|
| Baseline | 64 | 62.0 | 43.4 | 58.2 | 62.1 |
| 1min-3min | 64 | 59.8 | 43.1 | 57.4 | 62.2 |
| 2min-30min | 64 | **61.6** | **43.5** | **58.5** | **64.1** |

The transition from the baseline (LLaDA-V (You et al., 2025)) to Stage 1 yields a marked improvement across all metrics, raising the average score from 51.2 to 55.4. This suggests that the initial temporal alignment on diverse short clips is crucial for equipping the model with fundamental dynamic perception capabilities, as evidenced by substantial gains on Video-MME (56.4 to 61.8) and MLVU$_{dev}$ (59.4 to 64.4). Subsequently, Stage 2, which involves scaling the temporal resolution to 64 frames and extending the context window, further enhances performance. Finally, the incorporation of long-form video data in Stage 3 results in the highest overall performance (Avg 57.9). Improvements are most pronounced on datasets requiring sustained reasoning over extended temporal contexts, such as Video-MME (62.1 to 64.1) and LVBench (43.4 to 44.7), confirming the necessity of the curriculum's final adaptation phase for long-context understanding.

*Table 7.* **Stage Performance Comparisons.** The rightmost column reports the post-training GPU-hour cost of each curriculum stage.

| Stage | #F | Video-MMMU | LongVideoBench | LVBench | EgoSchema | MVBench | MLVU$_{dev}$ | MLVU$_{test}$ | Video-MME | **Avg** | GPU Hours |
|-------|----|------------|----------------|---------|-----------|---------|------------|-------------|-----------|--------|-----------|
| Baseline | 32 | 43.3 | 58.6 | 36.4 | 57.9 | 53.1 | 59.4 | 44.1 | 56.4 | 51.2 | – |
| *1* | 32 | 46.0 | 59.5 | 42.6 | 60.5 | 59.2 | 64.4 | 49.5 | 61.8 | 55.4 | 1361.5 |
| *2* | 64 | 44.9 | **62.0** | 43.4 | 63.7 | 58.2 | 68.2 | 53.4 | 62.1 | 56.9 | 834.4 |
| *3* | 64 | **46.6** | 61.4 | **44.7** | **64.5** | **59.4** | **69.2** | **53.4** | **64.1** | **57.9** | 680.3 |
| | | | | | | | | | *Total GPU Hours* | | **2876.2** |

## F.3. Main Results Across Input Frame Counts

To examine whether the main comparison is explained by using more frames, we evaluate VidLaDA from 8 to 64 input frames and compare it with LLaVA-OneVision at the matched 32-frame setting. As shown in Table 8, VidLaDA at 16 frames already matches or exceeds LLaVA-OneVision at 32 frames on most benchmarks, and VidLaDA at 32 frames outperforms it across all reported Table 1 benchmarks.

*Table 8.* **Main Results Across Input Frame Counts.** VidLaDA is evaluated from 8 to 64 frames and compared with LLaVA-OneVision at 32 frames.

| Model | #F | Video-MMMU | LongVideoBench | LVBench | EgoSchema | MVBench | MLVU$_{dev}$ | MLVU$_{test}$ | Video-MME |
|-------|----|------------|----------------|---------|-----------|---------|------------|-------------|-----------|
| LLaVA-OV | 32 | 33.9 | 56.5 | 40.7 | 60.1 | 56.7 | 64.7 | 45.3 | 58.5 |
| VidLaDA | 8 | 44.8 | 58.6 | 38.9 | 61.1 | 58.0 | 57.6 | 39.5 | 55.0 |
| VidLaDA | 16 | 46.1 | 60.1 | 40.4 | 62.6 | 59.2 | 61.4 | 45.3 | 57.5 |
| VidLaDA | 32 | 47.1 | 59.3 | 41.7 | 63.8 | 59.1 | 65.1 | 49.9 | 62.1 |
| VidLaDA | 64 | 46.6 | 61.4 | 44.7 | 64.5 | 59.4 | 69.2 | 53.4 | 64.1 |

## F.4. Peak Memory and TTFT

We additionally report practical efficiency measurements for the question-answer setting in Table 9. Peak memory for VidLaDA w/ MARS-Cache remains close to the AR baseline across tested output lengths, and model-side TTFT remains comparable to LLaVA-OneVision. Notably, unlike continuous diffusion generation, VidLaDA can produce readable text from the first decoding step rather than after multiple refinement iterations (Li et al., 2022).

## F.5. CoT Inference Visualizations

We present qualitative examples of VidLaDA's reasoning under the Chain-of-Thought framework, accelerated by MARS-Cache. Figure 8 displays the task, where the model utilizes intermediate temporal analysis to correctly reconstruct the

*Table 9.* **Practical Efficiency Measurements.** Peak VRAM is reported as midpoint $\pm$ half-range in GB, and model-side TTFT is measured in seconds under different generation lengths.

| Model | 1 token | | 32 tokens | | 128 tokens | |
|---|---|---|---|---|---|---|
| | VRAM | TTFT | VRAM | TTFT | VRAM | TTFT |
| LLaVA-OV | 21.6±0.2 | 0.5 | 21.6±0.2 | 0.5 | 21.6±0.2 | 0.5 |
| VidLaDA w/ MARS-Cache | 22.1±0.2 | 0.4 | 22.1±0.1 | 0.4 | 22.2±0.2 | 0.4 |

chronological order of disjoint scenes. Figure 9 illustrates the scenario, where the model successfully deduces the subject's expertise level by linking specific visual adjustments to causal intent.

### F.6. Architecture Comparison under Matched Fine-tuning

To disentangle architectural effects from training-related factors, we conduct a controlled fine-tuning experiment. Since re-training LLM and MLLM from scratch on the large-scale corpus and the visual instruction dataset is computationally prohibitive (Cheng et al., 2025), we compare two publicly available MLLM baselines with similar starting performance, i.e., LLaVA-OneVision as the AR baseline and LLaDA-V as the DLM baseline. We fine-tune both models using the same 10K samples and identical hyperparameters. Specifically, we freeze the ViT encoder and set the learning rate to 1e-5 for both the LLM and the MLP projector, other training settings follow Table 13. This protocol reduces differences stemming from data and optimization recipes, making the observed deltas more attributable to architectural choices.

The main matched fine-tuning results are reported in Table 3. The AR baseline exhibits a trade-off after one epoch and degrades overall after four epochs, whereas the DLM baseline improves more consistently and remains stable under matched fine-tuning. Under matched conditions, these results suggest that the DLM architecture may utilize the available spatiotemporal evidence more effectively than the AR counterpart, which is consistent with our understanding efficiency analysis in Proposition 3.2.

### F.7. Detailed Ablation Studies of MARS-Cache

In this section, we provide a comprehensive analysis of the proposed MARS-Cache. We validate our design choices regarding anchor token searching, modality-wise, and layer-wise refresh rates. Unless otherwise stated, all ablation experiments are conducted on EgoSchema (subset) and MLVU (test) to evaluate reasoning and long-context capabilities, respectively. The model layers are simply divided into four groups: Group 1 (layers 0-7), Group 2 (8-15), Group 3 (16-23), and Group 4 (24-31). The detailed settings for the ablation study are provided in Appendix G.5.

#### F.7.1. EFFECTIVENESS OF ANCHOR TOKEN SEARCHING

We first investigate the necessity of preserving global connectivity via anchor tokens and identify which network depths benefit most from this mechanism. We compare our anchor token searching against the baseline with full attention ("-") and variants where anchor tokens are removed (i.e., "0", frame-wise chunk attention).

Table 10 validates the necessity of anchor tokens, as removing them completely (Row 2) significantly degrades performance compared to the baseline. Regarding token allocation, we find that a tapered strategy assigning more tokens to shallow layers and fewer to deep layers yields optimal results (Last Row). Furthermore, maintaining full attention in shallow layers to preserve structural features while applying aggressive anchor pruning in deep layers (Row 4) achieves competitive performance with similar FLOPs. This suggests that deep semantic layers can be heavily compressed provided that early structural information remains intact.

#### F.7.2. MODALITY-WISE REFRESHING STRATEGY

A main premise of MARS-Cache is that visual representations are temporally more stable than textual hidden states during the decoding process. Consequently, the visual cache can be refreshed less frequently. We define the Vision/Text Refresh Ratio ($R_{v/t}$) as the ratio of their respective update intervals.

Figure 7 presents the results under baseline text refresh intervals ($\tau_{(*,t)} = 16$). We observe a consistent trend: increasing the refresh interval for vision tokens (i.e., $R_{v/t} > 1$) maintains or even slightly degrades accuracy while boosting Throughput

*Table 10.* **Ablation on Count of Anchor Token.** We compare the retention of anchor tokens across different layer groups. "-" indicates full attention (baseline), "0" indicates no anchor tokens (pure chunk attention), and numbers indicate the count of retained anchor tokens. FLOPs refers to the FLOPs of attention calculation only.

| G1 | G2 | G3 | G4 | Ego | MLVU | FLOPs |
|---|---|---|---|---|---|---|
| - | - | - | - | 65.0 | 42.6 | 100% |
| 0 | 0 | 0 | 0 | 63.8 | 40.7 | 53% |
| 32 | 32 | 32 | 32 | 65.0 | 41.5 | 66% |
| - | - | 32 | 32 | 65.2 | **45.3** | 83% |
| 128 | 128 | 128 | 128 | 64.6 | 44.4 | 93% |
| 128 | 32 | 32 | 32 | 65.6 | 41.8 | 73% |
| - | 32 | 32 | 32 | 65.8 | 41.7 | 75% |
| - | 128 | 64 | 32 | **66.8** | 44.0 | 84% |

(TPS). Specifically, with $\tau_{(*,t)} = 16$, the ratio of around 2 achieves the optimal balance. Pushing the ratio too high (e.g., $R_{v/t} > 2$) eventually degrades performance, indicating that while visual states drift slowly, they are not static.

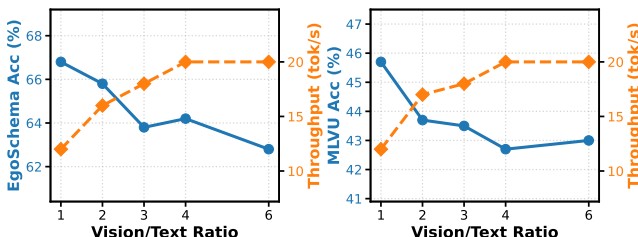

*Figure 7.* **Impact of Vision/Text Refresh Ratios.** We test varying the visual cache refresh interval while keeping the text interval fixed at 16.

### F.7.3. LAYER-WISE REFRESHING STRATEGY

Based on the observation that deep layers exhibit higher hidden state drift than shallow layers, we can utilize a hierarchical schedule where update frequencies increase with network depth.

Table 11 presents the results under this strategy. The results demonstrate that the pyramid refresh schedule (decreasing intervals from Group 1 to Group 4) yields the best performance. The configuration $64 \rightarrow 32 \rightarrow 16 \rightarrow 8$ (Row 6) achieves a high score of 68.0 on EgoSchema, and the second-best score 45.3 on MLVU, outperforming uniform schedules. Furthermore, when combining this with the Modality-wise strategy (where $R_{v/t} = 2$), we see further robustness. This confirms that the two refresh strategies are orthogonally efficient.

### F.7.4. OVERHEAD OF ANCHOR TOKEN SEARCHING

We analyze the overhead introduced by the adaptive anchor token searching mechanism. Specifically, we evaluate the impact of the number of sampled query tokens used to compute the proxy attention matrix. As shown in Table 12, using a very small number of tokens (16) for vision and text leads to unstable anchor token identification and lower performance. Conversely, using the full sequence to compute the attention score map incurs heavy computational and memory costs (OOM). We find that setting the query set size to 128 provides a robust trade-off between identifying high-quality global anchor tokens and maintaining low search overhead.

### F.8. Causal Inference Visualizations

To explore the potential of DLM-based Video LLMs in modeling complex event dependencies, we conduct a preliminary visualization case study following the methodology of (Chen et al., 2024b). As illustrated in Figure 10, the DLM-based Video LLM, VidLaDA, demonstrates the capability to identify non-local causal links (green arrows) bridging the temporal

*Table 11.* **Ablation on Layer-wise Refreshing.** We compare various hierarchical schedules. The cell background color indicates the update frequency: light green denotes low-frequency updates (e.g., interval 64), while **dark green** denotes high-frequency updates (e.g., interval 4/8).

| G1 | G2 | G3 | G4 | Ego | TPS | MLVU | TPS |
|----|----|----|----|-----|-----|------|-----|
| \multicolumn{8}{c}{*Uniform Modality Refresh* ($\tau_{(g,v)} = \tau_{(g,t)}$)} | | | | | | | |
| 32 | 32 | 32 | 32 | 65.0 | 16.2 | 42.6 | 16.7 |
| 64 | 64 | 32 | 32 | 64.8 | 18.4 | 41.2 | 18.5 |
| 32 | 32 | 16 | 16 | 66.6 | 13.9 | 44.5 | 14.0 |
| 32 | 32 | 16 | 8 | 67.0 | 12.0 | **45.9** | 12.2 |
| 32 | 16 | 8 | 4 | 66.6 | 8.2 | 44.8 | 8.3 |
| 64 | 32 | 16 | 8 | **68.0** | 12.5 | 45.3 | 12.6 |
| \multicolumn{8}{c}{*Modality-Aware Refresh* ($\tau_{(g,v)} = 2\tau_{(g,t)}$)} | | | | | | | |
| 32 | 32 | 16 | 8 | 67.0 | 12.0 | 45.9 | 12.2 |
| \multicolumn{4}{c}{($\times 2$)} | 66.8 | 16.4 | **46.9** | 16.7 |
| 32 | 16 | 8 | 4 | 66.6 | 8.2 | 44.8 | 8.3 |
| \multicolumn{4}{c}{($\times 2$)} | 66.4 | 12.2 | 44.3 | 12.4 |
| 64 | 32 | 16 | 8 | **68.0** | 12.5 | 45.3 | 12.6 |
| \multicolumn{4}{c}{($\times 2$)} | 65.8 | 16.9 | 45.3 | 17.0 |

*Table 12.* **Analysis of Search Token Overhead.** We measure the impact of the number of query tokens used for proxy attention on model performance and computational cost. GFLOPs and Memory indicate the overhead for a single calculation.

| #Vis | #Txt | Ego | MLVU | GFLOPs | Memory(MB) |
|------|------|-----|------|--------|------------|
| 16 | 16 | 65.0 | 42.6 | 7.1 | 82.6 |
| 32 | 32 | 64.8 | 43.9 | 14.1 | 165.2 |
| 64 | 64 | 64.2 | 44.0 | 28.2 | 330.3 |
| 128 | 128 | 65.2 | **45.1** | 56.4 | 660.6 |
| 256 | 256 | **66.6** | 44.0 | 112.7 | 1351.7 |
| $|V|$ | $|T|$ | - | - | - | **OOM** |

gap, whereas the AR baseline (LLaVA-OneVision) misses these dependencies. This demonstrates that DLM-based Video LLMs, leveraging their inherent bidirectional modeling capabilities, may possess significant potential to achieve robust performance in complex causal inference tasks.

# G. Experimental Details

## G.1. Experimental Setup

**Implementation Details.** VidLaDA is implemented using the LLaDA-8B (Nie et al., 2025) as the LLM backbone, coupled with SigLIP2-SO400M (Tschannen et al., 2025) as the ViT, following the LLaDA-V (You et al., 2025). The model is trained following the three-stage curriculum strategy described in Appendix E.1. The specific training settings are listed in Table 13. Training is conducted on NVIDIA H200 GPUs with a global batch size of 64. The learning rate can refer to Table 5.

**Datasets and Evaluation Benchmarks.** We conduct a comprehensive evaluation across eight diverse benchmarks to assess VidLaDA's general capabilities in video understanding. First, to evaluate holistic understanding across varied contexts and durations, we employ Video-MME (Fu et al., 2025) (w/o subtitles), MLVU (Zhou et al., 2025) (reporting both Dev and Test splits), LVBench (Wang et al., 2025c), and LongVideoBench (Wu et al., 2024). These benchmarks challenge models with complex visual information and sustained contexts, assessing the model's robustness in handling diverse video content ranging from short clips to extended sequences. Second, for fine-grained temporal perception, we utilize MVBench (Li et al., 2024b), which comprises 20 distinct tasks requiring precise action and attribute recognition. Third, we evaluate high-level

*Table 13.* **Training Setting Details for VidLaDA.**

| Item | Value |
|---|---|
| Total Batch Size | 64 |
| Warmup Ratio | 3% |
| Optimizer | AdamW |
| LR Scheduler | Cosine |
| Adam $\beta_1$ | 0.9 |
| Adam $\beta_2$ | 0.999 |
| Adam $\epsilon$ | 1e-8 |
| Weight Decay | 0 |
| Gradient clipping | 1.0 |

reasoning and expert knowledge acquisition using EgoSchema (Mangalam et al., 2023), a benchmark for complex intent understanding, and Video-MMMU (Hu et al., 2025), which tests multi-disciplinary professional understanding.

### G.2. Intra-Frame Understanding on MME

To empirically validate the spatial robustness of VidLaDA compared to Autoregressive baselines, as discussed in Section 3.1.1, we conduct a controlled spatial permutation experiment on the MME benchmark (Fu et al., 2026). For quantitative evaluation, we report the MME Sum score, which aggregates performance across the perception and cognition subtasks.

We employ an information-guided shuffling protocol to test the sensitivity of the model to the position of semantic features. First, we compute the $\ell_2$-norm of the output features from the vision encoder for every patch, serving as a proxy for information density. We then identify the top-$k$ tokens with the highest norms, setting $k = 256$, to isolate the most semantically salient regions of the image. To manipulate the sequence order, we define a position ratio $r \in [0, 1]$. The input sequence is constructed such that the selected high-norm tokens are placed starting at index $\lfloor r \times |\boldsymbol{E}^{\mathcal{V}}| \rfloor$, where $|\boldsymbol{E}^{\mathcal{V}}|$ is the total number of visual tokens. Consequently, $r = 0$ positions the high-information tokens at the visual start, while $r = 1$ pushes them to the visual end, with the remaining low-norm tokens filling the rest of the sequence. By evaluating the performance variance across different $r$ values, we can determine whether a model exhibits position-dependent early receptive field biases.

### G.3. Inter-Frame Interaction on RexTime

We evaluate variable temporal position reasoning using the ReXTime benchmark (Chen et al., 2024a), utilizing videos sourced from the QVHighlights (Lei et al., 2021) dataset. We conduct two distinct experiments to analyze the impact of event temporal position and sparsity in Section 3.1.2.

To analyze how the position of a critical event within a video affects model performance (corresponding to Figure 2b), we categorize the test samples based on the normalized temporal timestamp of the event. Let $T_V$ be the total duration of the video and $[t_{\text{start}}, t_{\text{end}}]$ be the ground-truth time interval of the relevant event. We define the *Event Position Ratio* as $r = (t_{\text{start}} + t_{\text{end}})/(2 \cdot T_V)$. We partition the test set into 15 uniform bins. We evaluate the model's accuracy within each bin using a uniform frame sampling strategy with $\{8, 32, 64\}$ frames. This setup reveals potential biases in processing information located at different temporal stages (start, middle, or end) of the sequence.

For the sparsity analysis (corresponding to Figure 2c), we examine the model's robustness to sparse visual evidence by progressively reducing the number of sampled key frames ($N_{\text{key}}$). Specifically, we vary $N_{\text{key}}$ in decreasing order (from 16 down to 1) to observe performance stability under constrained evidence. Furthermore, we introduce a variable number of non-key context frames ($N_{\text{noise}}$), for every fixed $N_{\text{key}}$, we iterate through a range of noise levels where the number of non-key frames is sampled from the set $\{0, 2, 4, \ldots, 32\}$. The final accuracy reported for a specific $N_{\text{key}}$ is the mean accuracy calculated across all these $N_{\text{noise}}$ settings, with the standard deviation illustrated as the shaded region in the figure.

### G.4. CoT Inference Settings

This section details the inference hyperparameters employed for the reasoning experiments using the Chain-of-Thought (CoT) prompts described in Appendix E.2. Regarding visual input processing, we uniformly sample up to 32 frames from the input video. For MARS-Cache settings, we adopt the optimal hyperparameters identified in the ablation studies (Section 5.3).

For the intermediate reasoning analysis, we allocate a generation budget of 1024 tokens for LongVideoBench and the EgoSchema (subset), and 2048 tokens for MLVU (test), to accommodate detailed thought processes, and we report the best performance achieved. Furthermore, capitalizing on the non-autoregressive nature of the underlying Diffusion Language Model, we enable parallel decoding (Wu et al., 2025) to accelerate the inference of these extended reasoning chains. Additionally, for the MLVU benchmark and LLaDA-V, we also employ CreditDecoding (Wang et al., 2025a) to ensure decoding stability.

Specifically, we adopt left-to-right block-wise decoding, a prevalent technique in current DLM inference that generates multiple tokens simultaneously within a sliding window (Nie et al., 2025; Ye et al., 2025c; You et al., 2025), and configure the process with a block length of 64.

### G.5. Ablation Settings

Distinct from the multi-stage Chain-of-Thought framework described in Appendix G.4, our ablation studies prioritize experimental control and throughput stability to rigorously evaluate the efficiency of the proposed MARS-Cache. To decouple the efficiency metrics from the variance introduced by dynamic task routing and variable-length generation, we adopt an explicit thought-tagging strategy inspired by LLaVA-CoT (Xu et al., 2025).

Specifically, we utilize a single-turn prompt design that instructs the model to enclose its reasoning process within specific tokens. Leveraging the unique capability of Diffusion Language Models to manipulate the target sequence initialization, we explicitly inject these start and end thinking tags into the masked sequence at predefined positions during the forward process. This constraint strictly enforces a fixed thinking generation length, thereby eliminating length-induced latency variations and ensuring that the reported Tokens Per Second (TPS) metrics reflect the pure algorithmic efficiency of the decoding strategy. Under this controlled protocol, we measure efficiency in terms of FLOPs and Throughput (TPS) on a single NVIDIA H200 GPU, configured with a generation length of 128, 128 diffusion steps, and a block length of 32, without employing parallel decoding.

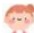 *In the video, a blonde girl appears, standing high up with a rose in her hand, overlooking the city. Then, a shirtless man wearing black shorts stands in front of a woman dressed in a bikini. There also appears a segment with three people on a boat traveling on the sea* **What is the order of these scenes?**

**A**. *First, a blonde girl appears, standing high up with a rose in her hand, overlooking the city. Then, a shirtless man wearing black shorts stands in front of a woman dressed in a bikini, and a segment with three people on a boat traveling on the sea appears.*
**B**. *First, a shirtless man wearing black shorts stands in front of a woman dressed in a bikini. Then, a blonde girl appears, standing high up with a rose in her hand, overlooking the city. Finally, a segment with three people on a boat traveling on the sea appears.*
**C**. *First, a segment with three people on a boat traveling on the sea appears. Then, a shirtless man wearing black shorts stands in front of a woman dressed in a bikini. Finally, a blonde girl appears, standing high up with a rose in her hand, overlooking the city.*
**D**. *First, a **blonde girl** appears, standing high up with a rose in her hand, overlooking the city. Then, a **shirtless man wearing black shorts** stands in front of a woman dressed in a bikini. Finally, a **segment with three people** on a boat traveling on the sea appears.*

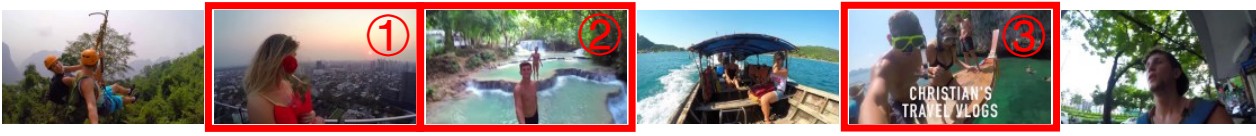

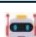 → *Dynamic Action & Temporal Evolution*

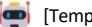 [Temporal Timeline]:
1. **Scope Definition**: A ***blonde girl*** standing high up with a rose in her hand, overlooking the city.
2. **Action Segmentation**: The scene starts with a blonde girl standing high up with a rose in her hand, overlooking the city.
3. **Dynamics Analysis**: The girl is standing at a high elevation, with a panoramic view of the city.
4. **Sequence Verification**: *This is the **first** event in the sequence.*

5. **Scope Definition**: A ***shirtless man*** wearing black shorts standing in front of a woman dressed in a bikini.
6. **Action Segmentation**: The scene transitions to a shirtless man wearing black shorts standing in front of a woman dressed in a bikini.
7. **Dynamics Analysis**: The man is standing in front of the woman, with no significant movement depicted in the scene.
8. **Sequence Verification**: *This is the **second** event in the sequence.*

9. **Scope Definition**: ***Three people on a boat*** traveling on the sea.
10. **Action Segmentation**: The final segment shows three people on a boat traveling on the sea.
11. **Dynamics Analysis**: The boat is moving through the water, and the people are engaged in the activity.
12. **Sequence Verification**: *This is the **third** event in the sequence.*

*The order of these scenes is: the blonde girl standing high up with a rose in her hand, the shirtless man in black shorts standing in front of a woman in a bikini, and the three people on a boat traveling on the sea.*

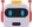 *The Answer is:* **D**.

*Figure 8.* **Reasoning Analysis.** An inference example of VidLaDA via MARS-Cache on a case of *Dynamic Action & Temporal Evolution*.

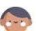 *What can be deduced about c's level of expertise in the task by **observing the kind of adjustments made throughout the video**?*

**A**. C is a novice woodworker. he was not able to cut the wood to size and install it on the wall without making several adjustments.
**B**. C is an expert woodworker. he was able to cut the wood to size and install it on the wall without making any adjustments.
**C**. C is a professional woodworker. he was able to cut the wood to size and install it on the wall in a timely and efficient manner.
**D**. C is an **experienced** woodworker. he was able to **cut the wood** to size and install it on the wall **with few adjustments**.
**E**. C is an amateur woodworker. he was able to cut the wood to size and install it on the wall, but he took a long time to do so.

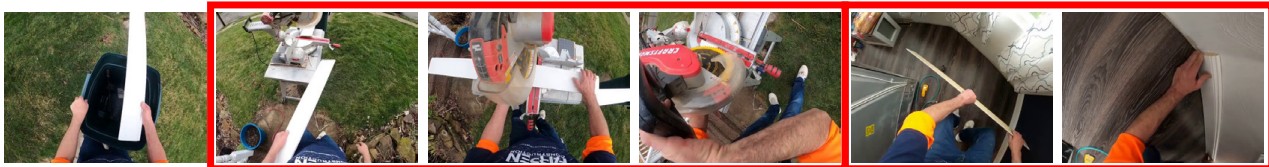

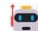 → *Complex Logic & Causal Reasoning*

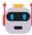 1. **Question Analysis**: The question aims to deduce the level of expertise of the person "c" in the task by observing the adjustments made throughout the video.
2. **Evidence Linking**: The person is seen handling various tools and materials, including a circular saw, a table saw, and a measuring tape. They make precise adjustments to a white strip of material, indicating a high level of care.
3. **Causal Inference**: The meticulous use of tools and the careful handling of the material suggest that the person has a high level of **expertise** in the task.
4. **Contextual Exclusion**: **The video does not depict any rushed or careless actions**, which would indicate a lower level of expertise. Instead, the person demonstrates detailed and careful attention to the task.

[Reasoning Chain]: **The person demonstrates a **high level of expertise** in the task, as evidenced by their meticulous use of tools and careful handling of the material.**

The Answer is: **D**.

*Figure 9.* **Reasoning Analysis.** An inference example of VidLaDA via MARS-Cache on a case of *Complex Logic & Causal Reasoning*.

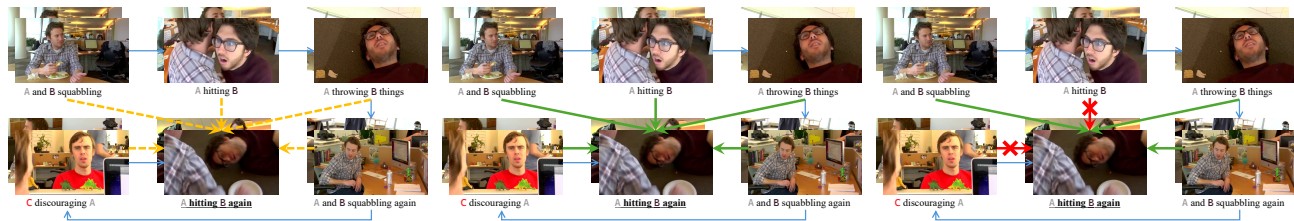

*(a)* Ground Truth Causal Graph.   *(b)* Causal Inference from VidLaDA.   *(c)* Causal Inference from LLaVA-OV.

*Figure 10.* **Causal Inference Examples.** The blue arrows indicate the chronological video order. Dashed yellow arrows denote the ground truth causal dependencies. Green arrows represent correctly predicted links, while red arrows and crosses mark missed dependencies. VidLaDA successfully models long-range causal links, whereas the AR baseline fails to connect distant events.

*Table 14.* **The Prompt Template for Stage 1: Task Prompt Routing**. The model classifies the user query into one of six distinct reasoning domains.

---

### Stage 1: Task Prompt Routing
**ROUTER_TEMPLATE** = """"
# Role
You are an expert Task Router for a Video Understanding AI System.
Your objective is to select the single best reasoning strategy (Category) for the User's Question.
# Input Data
- Video Duration: {duration}
- User Question: "{question}"
# Evaluation Criteria (The Strategies)
1. **Static Entity & Spatial Perception**:
  - Focus: Static visual details in a single frame (color, object count, OCR text, relative position).
  - Use when: The question is about ˍvisible propertiesˍ regardless of time.
2. **Dynamic Action & Temporal Evolution**:
  - Focus: Movements, action sequences, speed, trajectories, or changes over time.
  - Use when: The question contains dynamic verbs or asks about the ˍflow of eventsˍ.
3. **Complex Logic & Causal Reasoning**:
  - Focus: Inferred meaning, intentions, "Why", emotions, social interactions, or counterfactuals.
  - Use when: The answer is not explicitly visible and requires deep thinking.
4. **Fine-grained Retrieval & Temporal Grounding**:
  - Focus: Locating a specific timestamp, frame, or "needle" in a haystack.
  - Use when: The question asks "When", "Find the moment", or for a specific detail in a long video.
5. **Global Summarization & Holistic Understanding**:
  - Focus: Summaries, titles, main themes, or condensing the video content.
  - Use when: The user wants a high-level overview.
6. **General / Uncertainty / Fallback (Default Mode)**:
  - Focus: Conversational queries, simple greetings, vague questions, or **when you are UNSURE**.
  - Use when:
      1. The question does not clearly fit 1, 2, 3, 4, or 5.
      2. The question is ambiguous.
      3. You feel low confidence in assigning a specialized category.
  - **Strategy**: It is better to choose 6 than to force a wrong specialized category.
# Decision Logic & Rules
1. Analyze the question's intent against the keywords of 1-5.
2. **The "Uncertainty Rule"**: If the question seems to straddle multiple categories significantly, or if it lacks specific keywords to trigger 1-5, **IMMEDIATELY CHOOSE 6**.
3. Do not over-interpret simple questions. For example, "What is this?" is typically 6 (General), unless it asks "What action is this?" (2).
# Output
Return ONLY the single integer of the category (1, 2, 3, 4, 5, or 6).
""""

---

*Table 15.* **Reasoning Analysis Prompts (Part I)**. These prompts are selected if the Router outputs Category 1, 2, or 3, focusing on static perception, dynamic action, or complex logic, respectively.

---

### Stage 2: Reasoning Analysis (Visual, Temporal, Logic)
**PROMPT_STATIC_SPATIAL_ANALYSIS** = """"
# Role
You are a Computer Vision Specialist focusing on fine-grained visual details.
# Input
User Question: "{question}"
# Task
Based strictly on the User Question above, perform a "Visual Scan" to extract relevant visual evidence.
**Focus ONLY on the specific objects, attributes, or text mentioned or implied by the question.** DO NOT answer the question directly yet.
# Visual Scan Instructions
1. **Target Identification**: Identify the key objects or regions specifically requested in the question.
2. **Attribute Verification**: Analyze the color, shape, count, or text (OCR) of these specific targets.
3. **Spatial Check**: Describe the relative position of these targets if relevant to the question.
4. **Filtering**: Ignore visual details that are irrelevant to the user's specific inquiry.
# Output Format
[Visual Scan]: <Provide detailed observations strictly related to the question targets.>"""

---

**PROMPT_DYNAMIC_TEMPORAL_ANALYSIS** = """"
# Role
You are an Action Analysis Expert.
# Input
User Question: "{question}"
# Task
Construct a "Temporal Timeline" for the specific actions or events queried by the user.
**Filter out background activities that do not help answer the specific question.**
DO NOT answer the question directly yet.
# Timeline Instructions
1. **Scope Definition**: Identify which specific actor or event the question is asking about.
2. **Action Segmentation**: Log the start and end of only the relevant actions.
3. **Dynamics Analysis**: Describe the speed, direction, or intensity specifically for the queried action.
4. **Sequence Verification**: Ensure the chronological order matches the events mentioned in the question.
# Output Format
[Temporal Timeline]: <Provide a step-by-step breakdown of the relevant actions.>"""

---

**PROMPT_COMPLEX_LOGIC_ANALYSIS** = """"
# Role
You are a Cognitive Analyst.
# Input
User Question: "{question}"
# Task
Generate a "Reasoning Chain" to bridge the visual gap for the specific question.
**Your goal is to find the "Why" or "How" behind the specific subject raised in the question.**
DO NOT answer the question directly yet.
# Reasoning Chain instructions
1. **Question Analysis**: What is the core mystery or implicit relationship in the question?
2. **Evidence Linking**: Connect specific visual cues (expressions, items) directly to the question's intent.
3. **Causal Inference**: Deduce the cause or motivation specifically explaining the queried event.
4. **Contextual Exclusion**: Discard reasoning paths that do not help solve the specific question.
# Output Format
[Reasoning Chain]: <Provide step-by-step logical deductions leading towards the answer.>"""

*Table 16.* **Reasoning Analysis Prompts (Part II)**. These prompts are selected if the Router outputs Category 4, 5, or 6, handling specific retrieval, summarization, or general fallback scenarios.

---

### Stage 2: Reasoning Analysis (Retrieval, Summarization, Fallback)
**PROMPT_RETRIEVAL_GROUNDING_ANALYSIS** = """
# Role
You are a Precision Video Retrieval Assistant.
# Input
User Question: "{question}"
# Task
Locate the specific moment or detail requested in the question.
**Your search must be strictly limited to the criteria defined in the question.**
DO NOT answer the question directly yet.
# Retrieval Strategy
1. **Keyword Mapping**: Map the question's keywords to visual or auditory signatures (the "Target").
2. **Focused Scanning**: Scan the video specifically looking for these signatures.
3. **Timestamping**: Pinpoint the exact moments where the queried event occurs.
4. **Match Verification**: Confirm that the found segment perfectly matches the question's description.
# Output Format
[Target Details]: <Describe the found evidence and its specific location/time.>"""

---

**PROMPT_GLOBAL_SUMMARIZATION_ANALYSIS** = """
# Role
You are a Professional Content Editor.
# Input
User Question: "{question}"
# Task
Extract "Synopsis Elements" necessary to satisfy the user's specific request.
**Note: Do not just summarize the whole video. Summarize the aspect requested by the user.**
DO NOT answer the question directly yet.
# Synopsis Generation
1. **Topic Alignment**: Determine which aspect of the video (plot, theme, character arc) the user is asking about.
2. **Key Moment Extraction**: Select only the turning points that contribute to answering the question.
3. **Flow Description**: Describe the progression (Intro to Conclusion) relative to the user's topic.
4. **Noise Reduction**: Omit plot details that are unrelated to the specific question.
# Output Format
[Global Synopsis]: <Provide structured content elements focused on the query.>"""

---

**PROMPT_GENERAL_FALLBACK_ANALYSIS** = """
# Role
You are a helpful AI Video Assistant.
# Input
User Question: "{question}"
# Task
Provide a set of "General Observations" that will help answer the specific question above.
**Focus your observation on the entities or concepts mentioned in the question.**
DO NOT answer the question directly yet.
# Input
Question: {question}
# Output Format
[Observations]: <List key visual details strictly relevant to answering the question.>"""

*Table 17.* **Prompts for Self-Reflection (Stage 3) and Final Answer Generation (Stage 4)**, followed by the overall logic of the inference pipeline. The reflection step ensures that only high-quality intermediate reasoning is used for the final response.

---

### Stage 3: Self-Reflection
**PROMPT_REFLECTION_BOOLEAN** = """"
# Role
You are a Quality Assurance Filter.
# Task
According to the video, evaluate if the provided **Analysis** is relevant, consistent, and helpful to answer the **User Question**.
# Input
1. User Question: "{question}"
2. Analysis: "{analysis_text}"
# Evaluation Rules
- Return **YES** if the analysis is specific, relevant, and logically sound.
- Return **NO** if the analysis is irrelevant, hallucinatory, contradictory, or empty.
# Output Constraint
Output ONLY the word "YES" or "NO". Do not answer the question, or provide any explanation, or markdown formatting.
""""

### Stage 4: Final Answer Generation
**PROMPT_FINAL_WITH_CONTEXT** = """"
# Task
Answer the user's question based on the provided **Video** and the **Expert Analysis** below.
# Instructions
1. **Synthesize**: Use the "Expert Analysis" as a reliable guide to interpret the video content. It highlights the key moments or details relevant to the question.
2. **Answer**: Provide a direct, clear, and natural answer to the question.
3. **Refinement**: Do not explicitly say "Based on the analysis..." or "The expert mentioned...". Just answer the question naturally as if you observed these details yourself.
# Input Data
1. **Expert Analysis (Context)**:
"{analysis_text}"
2. **User Question**: "{question}"
# Output Format
[Answer]: <Your final, concise answer>
""""

---

# Inference Pseudo-Code
category = model.generate(ROUTER_TEMPLATE.format(question=user_q))
expert_prompt = PROMPT_MAP[category]
analysis = model.generate(expert_prompt.format(question=user_q))
is_valid = model.generate(PROMPT_REFLECTION_BOOLEAN.format(analysis=analysis))
final_ans = Final_Gen(model, analysis, is_valid, PROMPT_FINAL_WITH_CONTEXT)

