# OpenReview forum: "VidLaDA: Bidirectional Diffusion Large Language Models for Efficient Video Understanding"
_ICML.cc/2026/Conference — ICML 2026 regular_

### Official Review · Reviewer_J9EC · 2026-03-12

**Soundness:** 3
**Presentation:** 3
**Significance:** 2
**Originality:** 2
**Overall Recommendation:** 5
**Confidence:** 4

**Summary:**

This paper proposed a diffusion based video LLM that replaces autoregressive models. It aims to tackle the problem that casual attention in autoregressive models creates asymmetric relations between video tokens, which harms temporal reasoning. Therefore, the method uses bideirectional attention based diffusion in order to paralelly predict the tokens, to improve reasoning. Furthermore, the paper introduces MARS-Cache, in order to tackle the compute burden of diffusion decoding.

**Compliance With Llm Reviewing Policy:**

Affirmed.

**Final Justification:**

The rebuttal addressed all my concerns and I want to update the score.

**Key Questions For Authors:**

I would like the author to answer the following:

1. Can you provide an ablation isolating the effect of bidirectional decoding itself, independent of other architectural or training differences?
2. Can you provide comparison describing memory usage for the various methods?

**Limitations:**

As I have previously mentioned a couple of times above, one clear limitation is the unclear attribution of bidirectional decoding to the improvement. The claim that bidirectional diffusion decoding is the key improvement makes sense, but the experiment partially support it. It could also be that the main improvement comes from the training methodology itself, rather than the architecture.

The other limitation is still the complexity of the overall system. Diffusion decoding remains computationally demanding even with the proposed caching mechanism. Together with the absence of memory analysis, it's not clear if the overall complexity reduces.

**Strengths And Weaknesses:**

Strengths:

1. The paper claims to be the first to propose a diffusion-basesd video LLM with biderctional attentin, which itself sounds innovative. It presents it clearly and justifies the design choices very reasonably.
2. The paper also supplies reasonable theoretical motivation, that asymmetric receptive fields limit information capacity and video autoregressive models.
3. The paper further tackles its own introduced limitation, which is the compute burden of diffusion models. It does so via MARS-Cache.

Weaknesses:
1. Given the theoretical justification, it's still not clear if this is a practical limitation. The experiment rather compares autoregressive models and diffusion models, but doesn't support the claim that bidirectional generation is the key to the improvement.
2. Despite the MARS-Cache, the method still remains computationally demanding, with requiring many denoising steps, which limits its usage in practice. Aside from runtime, what about memory usage?

---

> ### Author Rebuttal · Authors · 2026-03-30
>
> Thank you for your valuable feedback. The reviewer's remarks are in *italics*, followed by our responses.
>
>
>
> > *W1 & Q1 - Bidirectional decoding is not clearly shown to be the key improvement. Can you isolate it from training differences?*
>
>
>
> To clarify this point, we provide the following three complementary pieces of evidence:
>
> (1) Section 3.1 is not only theoretical. Figure 2 already provides controlled diagnostics: when salient tokens are moved spatially, AR degrades while DLM stays stable; when evidence appears later or becomes sparse, the DLM remains much more stable (Section 3.1).
>
> (2) Appendix F.4 reduces training confounds directly. It compares two publicly available MLLMs with similar starting performance and fine-tunes them with the same 10K data and identical hyperparameters, because re-training full LLM+MLLM pipelines from scratch is computationally prohibitive (Appendix F.4). To further reduce bias from incomplete data fitting, we extended this matched setting from 1 epoch to 4 epochs:
>
> |**Model**|**#F**|**LongVideoBench**|**LVBench**|**MVBench**|**Video-MME**|**Avg**|
> |---|---|---|---|---|---|---|
> |LLaVA-OV|32|56.5|40.7|56.7|58.5|53.1|
> |+SFT 1 Epoch|32|57.4 (+0.9)|38.8 (-1.9)|54.1 (-2.6)|58.6 (+0.1)|52.2 (-0.9)|
> |+SFT 4 Epoch|32|56.0 (-0.5)|38.3 (-2.4)|54.5 (-2.2)|56.8 (-1.7)|51.4 (-1.7)|
> |LLaDA-V|32|58.2|36.4|53.1|56.4|51.0|
> |+SFT 1 Epoch|32|58.9 (+0.7)|36.6 (+0.2)|53.7 (+0.6)|58.5 (+2.1)|51.9 (+0.9)|
> |+SFT 4 Epoch|32|58.0 (-0.2)|37.1 (+0.7)|53.6 (+0.5)|58.7 (+2.3)|51.9 (+0.9)|
>
> Under matched fine-tuning, AR gains on LongVideoBench at 1 epoch but degrades overall by 4 epochs, whereas DLM remains stable and gains more on Video-MME.
>
> (3) We further run the Figure 2c sparse-evidence diagnostic using these matched checkpoints:
>
> | **Model**   | **Ep** | **Acc** |
> | ----------- | ------ | ------- |
> | Matched AR  | 1      | 83.8    |
> | Matched DLM | 1      | 88.3    |
> | Matched AR  | 4      | 81.6    |
> | Matched DLM | 4      | 87.2    |
>
> The DLM stays ahead by 4.5-5.6 points, making it less likely that the gain comes only from training recipe rather than architecture.
>
> > *W2 & Q2 - Computationally demanding with many denoising steps. Memory usage?*
>
>
>
> We first clarify that VidLaDA is a discrete DLM: at each decoding step, it updates multiple masked output positions in parallel, unlike continuous diffusion models that require many iterative refinement steps before producing readable text. By contrast, AR decoding generates one token per step. Thus, for outputs of the same length, VidLaDA can require fewer decoding steps than AR. For the practical setting, peak VRAM is:
>
> | **Gen Length** | **LLaVA-OV (GB)** | **VidLaDA (GB)** | **VidLaDA+MARS (GB)** |
> | -------------- | ----------------- | ---------------- | --------------------- |
> | 1 token        | 21.4-21.7         | 20.2-20.3        | 21.9-22.2             |
> | 32 tokens      | 21.4-21.7         | 23.7-23.8        | 22.0-22.2             |
> | 128 tokens     | 21.4-21.7         | 23.8-23.9        | 22.0-22.3             |
>
> In this practical setting, peak memory remains close to AR: LLaVA-OV stays at 21.4-21.7 GB, while VidLaDA+MARS stays around 21.9-22.3 GB across the tested output lengths.
>
> We further clarify that MARS-Cache does not refresh all visual states from scratch at every decoding step. Its primary mechanism is multimodal asynchronous refreshing: visual states are refreshed less frequently than text states, while deeper layers are refreshed more frequently (Section 4.2). As a result, many decoding steps reuse cached visual states instead of rerunning the full visual computation, and frame-wise chunk attention further reduces the update cost when refresh is required. Therefore, MARS-Cache substantially reduces the amount of visual computation needed for state updates, thereby reducing the computational demand in practice. This is also reflected in Table 2: with MARS-Cache, VidLaDA achieves higher throughput.
>
> We therefore do not think the current DLM overhead should be interpreted as showing that the method lacks practical value. A closer analogy is the early stage of AR long-context modeling: before dedicated systems optimizations, AR models also faced substantial memory and latency overhead, yet later engineering significantly narrowed those gaps. Video DLMs are still at a similarly early stage. Even so, our current results already show concrete practical relief: MARS-Cache reduces repeated visual computation and improves throughput. More broadly, ongoing DLM research is already continuing to improve architectural efficiency and decoding efficiency, while preserving the core bidirectional decoding advantages. Together with our work here, this helps pave the way for broader practical usage of Video DLMs. We will clarify this point more explicitly in the camera-ready discussion.

---

> > ### Author Rebuttal · Reviewer_J9EC · 2026-04-01
> >
> > **W1 & Q1 - Bidirectional decoding is not clearly shown to be the key improvement. Can you isolate it from training differences?**
> >
> > I knowledge the reasoning given in the rebuttal.
> >
> > **W2 & Q2 - Computationally demanding with many denoising steps. Memory usage?**
> >
> > I appreciate the broad explanation given in the rebuttal. I think it's important to add this explanation to the main text.
> >
> > My concerns are **fully solved** and I will update the score accordingly.

---

> > > ### Author Response · Authors · 2026-04-02
> > >
> > > Thank you for your thoughtful review and for updating your score. We are glad that our explanations have fully resolved your concerns. As you suggested, we will incorporate our discussion into the final revision. Once again, we extend our heartfelt gratitude for your positive feedback and invaluable contributions to enhancing the quality of our work.

---

### Official Review · Reviewer_c8SW · 2026-03-12

**Soundness:** 3
**Presentation:** 4
**Significance:** 3
**Originality:** 3
**Overall Recommendation:** 4
**Confidence:** 4

**Summary:**

This paper introduces VidLaDA, a novel Video LLM built on Diffusion Language Models (DLMs) to overcome the limitations of the traditional autoregressive paradigm, namely unidirectional spatiotemporal modeling and serial decoding latency. By employing bidirectional attention for global video understanding and parallel token generation, and further optimizing efficiency with the MARS-Cache strategy, VidLaDA achieves a performance comparable to state-of-the-art autoregressive models while delivering significantly higher inference throughput (up to 12.8x faster than vanilla DLM and surpassing optimized AR baselines).

**Compliance With Llm Reviewing Policy:**

Affirmed.

**Key Questions For Authors:**

Please refer to Weeknesses

**Limitations:**

No. While the authors briefly touch upon the scope of their model, the paper would benefit from a more rigorous discussion of (1) the increased peak VRAM requirements compared to AR models, (2) the inherent "Time to First Token" latency in diffusion decoding, and (3) the ethical implications of high-efficiency long-video analysis in surveillance contexts. Admitting these trade-offs would strengthen the transparency of the work.

**Strengths And Weaknesses:**

Strengths
- The paper successfully challenges the dominant AR paradigm by introducing a Video LLM based on DLMs. The use of bidirectional attention for video understanding is a highly creative approach that addresses the inherent "causal mask" limitation in traditional models.
- The proposed MARS-Cache is a well-designed technical contribution. By implementing Frame-wise Chunk Attention and Adaptive Anchor Token Searching, the authors effectively reduce the $O(N^2)$ complexity of bidirectional attention to $O(N)$, making DLMs practical for long-video tasks.
- The model's performance on comprehensive benchmarks (Video-MME, MVBench, LongVideoBench) is solid, showing that VidLaDA matches the state-of-the-art while offering superior global spatiotemporal reasoning capabilities.

Weaknesses
- While a 1.3x speedup over AR models is technically impressive, it may not be sufficient to justify the significantly higher Peak VRAM usage and computational intensity inherent in bidirectional diffusion. Bidirectional models require maintaining a dense attention matrix for all tokens during each denoising step, which could lead to a much higher "cost-per-token" in terms of energy and hardware memory compared to the streamlined KV-Cache in AR models.

- The paper lacks a detailed comparison of training costs. Diffusion-based language models generally require more training steps and data to achieve the same level of linguistic fluency and instruction-following capability as pre-trained AR models. The three-phase curriculum training seems intensive, and the total GPU-hour investment remains unclear.

- The "Multimodal Asynchronous Refreshing" strategy introduces specific refresh rates for vision and text tokens. The performance might be sensitive to these rates depending on video dynamics (e.g., fast-paced action vs. static slides). The paper would be stronger with an analysis of how these hyperparameters generalize across different video genres.

- In real-time interaction, the "Time to First Token" (TTFT) is critical. Diffusion models must complete several denoising iterations before any readable text is produced, whereas AR models output tokens progressively. This inherent latency characteristic of DLMs is not fully addressed in the efficiency analysis.

---

> ### Author Rebuttal · Authors · 2026-03-30
>
> Thank you for your valuable feedback. We provide detailed responses below. The reviewer's remarks are in *italics*, followed by our responses.
>
>
>
> > *W1 - VRAM overhead and cost-per-token of bidirectional diffusion vs. AR KV-Cache.*
>
> |**Gen Length**|**LLaVA-OV (GB)**|**VidLaDA (GB)**|**VidLaDA+MARS (GB)**|
> |---|---|---|---|
> |1 token|21.4-21.7|20.2-20.3|21.9-22.2|
> |32 tokens|21.4-21.7|23.7-23.8|22.0-22.2|
> |128 tokens|21.4-21.7|23.8-23.9|22.0-22.3|
>
> In this question-answer setting, peak memory remains close to AR: LLaVA-OV stays at 21.4-21.7 GB, while VidLaDA+MARS stays around 21.9-22.3 GB across the tested output lengths.
>
> Regarding the "dense attention matrix" concern, MARS-Cache does not refresh all visual states from scratch at every step. Its primary mechanism is multimodal asynchronous refreshing: visual states are refreshed less frequently than text states, while deeper layers are refreshed more frequently (Section 4.2). As a result, many decoding steps reuse cached visual states instead of rerunning the full visual computation, which also means they do not repeatedly recompute the full-attention dense attention matrix over all visual tokens. When refresh is required, frame-wise chunk attention further reduces the update cost. Therefore, MARS-Cache substantially reduces the amount of visual computation needed for state updates, thereby reducing the computational demand in practice. This is also reflected in Table 2: with MARS-Cache, VidLaDA achieves higher throughput.
>
>
>
> > *W2 - Training cost comparison and GPU-hour investment.*
>
> VidLaDA is built on a masking-based discrete DLM backbone and follows the same LLaDA-style training paradigm. The GPU-hour table below summarizes the three-stage video post-training cost used in this work:
>
> | **Stage** | **GPU Hours** |
> | --------- | ------------- |
> | Stage 1   | 1361.5        |
> | Stage 2   | 834.4         |
> | Stage 3   | 680.3         |
> | Total     | 2876.2        |
>
> The separate question is whether DLMs inherently require more supervised video data or post-training steps than AR models. Appendix F.4 directly addresses this point: the AR and DLM baselines are fine-tuned on the same 10K samples with identical hyperparameters, yet the DLM improves more consistently. Thus, in our setting, we do not observe evidence that the DLM requires more supervised steps than the AR baseline.
>
>
>
> > *W3 - Sensitivity of refresh rates to video dynamics and genre generalization.*
>
> First, the results in Table 2 are not obtained through benchmark-specific tuning of MARS-Cache for each evaluation setting. Since EgoSchema (subset), MLVU  (test), and LongVideoBench cover substantially different video characteristics and genres, MARS-Cache remaining effective across these benchmarks already supports generalization beyond a single video regime.
>
> To further probe sensitivity to video dynamics, we estimate a simple motion score for each EgoSchema (subset) video as follows: for each video, we first measure motion between adjacent sampled frames, then use an upper-quartile summary of these motion values as its dynamics score. We then split this video dataset into the lower and upper 50% of this score, i.e., low- and high-dynamics subsets:
>
> | **Method**         | **Overall** | **Low** | **High** | **Gap** |
> | ------------------ | ----------- | ------- | -------- | ------- |
> | VidLaDA w/o Cache  | 67.4        | 66.4    | 68.4     | 2.0     |
> | VidLaDA+Dual-Cache | 67.2        | 66.4    | 68.0     | 1.6     |
> | VidLaDA+MARS-Cache | 67.0        | 67.2    | 66.8     | 0.4     |
>
> VidLaDA+MARS-Cache shows the smallest low/high gap (0.4), whereas the w/o-cache and dual-cache variants exhibit larger motion-dependent variation. This suggests that the chosen MARS-Cache refresh setting is reasonably stable across different video dynamics.
>
>
>
> > *W4 - TTFT latency due to iterative denoising in diffusion models.*
>
>
>
> A key distinction is that VidLaDA is a discrete DLM, not a continuous diffusion model: each decoding step predicts masked tokens in parallel, so readable text is available from the first decoding step rather than only after many refinement iterations. Correspondingly, model-side TTFT remains low and changes only modestly across the tested output lengths:
>
> |**Gen Length**|**LLaVA-OV (s)**|**VidLaDA (s)**|**VidLaDA+MARS (s)**|
> |---|---|---|---|
> |1 token|0.5|0.5|0.4|
> |32 tokens|0.5|0.5|0.4|
> |128 tokens|0.5|0.5|0.4|
>
> > *Ethical implications.*
>
>
>
> We agree that this point should be discussed more explicitly. High-efficiency long-video analysis may raise privacy, consent, and surveillance-adjacent misuse concerns. We will address this in the camera-ready Impact Statement.

---

### Official Review · Reviewer_ajbR · 2026-03-13

**Soundness:** 4
**Presentation:** 4
**Significance:** 4
**Originality:** 3
**Overall Recommendation:** 5
**Confidence:** 4

**Summary:**

This paper proposes VidLaDA, a bidirectional Diffusion-based Video Large Language Model, to address inherent limitations in autoregressive (AR) models for video understanding tasks. The authors argue that AR models, constrained by causal attention, inherently bias early video tokens and inadequately integrate dispersed spatiotemporal evidence. By employing full bidirectional attention, VidLaDA demonstrates superior robustness in aggregating key visual information across spatial and temporal dimensions. Additionally, to mitigate the computational overhead typically associated with diffusion models, the authors introduce MARS-Cache, a structured caching mechanism that leverages visual input redundancy through asynchronous updates, frame-wise attention locality, and strategically placed anchor tokens. Experimental results validate VidLaDA's superior accuracy compared to AR baselines across various video benchmarks and demonstrate substantial speedups achieved by MARS-Cache.

**Compliance With Llm Reviewing Policy:**

Affirmed.

**Final Justification:**

I consider this work technically robust, clearly structured, and insightful. All of my concerns have been addressed. Therefore, I maintain my original positive score.

**Key Questions For Authors:**

1. The proposed model uses SigLIP-2, whereas the compared models (i.e., LLaVA-OneVision and LLaVA-Video) employ the original SigLIP. Could this discrepancy lead to an unfair comparison?

2. In Figure 3, I observe a distribution pattern in the Global Anchor Tokens that closely resembles the “attention sink” phenomenon reported in StreamingLLM [1], i.e., strong attention concentration on the first few tokens of the sequence. However, given that DLLMs employ bidirectional attention, softmax-induced persistent focus on early tokens should theoretically not occur. Could the authors clarify this contradiction?

3. The Global Anchor Tokens phenomenon appears highly related to the “temporal attention outliers” identified in DyToK (Appendix B.1) [2]. Although the model architectures differ, the underlying mechanism revealed in this paper may offer a plausible explanation for DyToK’s observations. The authors are encouraged to cite and analyze DyToK’s findings, which would enrich the paper’s insights and benefit future research.

[1] Efficient Streaming Language Models with Attention Sinks. ICLR 2024.

[2] Less Is More, but Where? Dynamic Token Compression via LLM-Guided Keyframe Prior. NeurIPS 2025.

**Limitations:**

yes

**Strengths And Weaknesses:**

### Strengths

1. **Soundness:** The proposed model architecture and acceleration methodology are technically robust and effectively validated through extensive experiments, demonstrating significant improvements over auto-regressive counterparts.

2. **Presentation:** The manuscript is clearly structured, and the overall narrative is easy to follow. Figures and diagrams are well-designed and aid understanding effectively.

3. **Significance:** This work addresses a highly relevant problem within video-language modeling and proposes innovative techniques that enhance the understanding of attention mechanisms in video diffusion language models. The findings are insightful and have broad implications for future research.

### Weaknesses

No major weaknesses are observed, but clarification on the following questions would strengthen the paper and may affect my final score.

---

> ### Author Rebuttal · Authors · 2026-03-30
>
> Thank you for the positive evaluation and insightful questions. The reviewer's remarks are in *italics*, followed by our responses.
>
> > *Q1 - SigLIP-2 vs SigLIP encoder discrepancy and fairness of comparison.*
>
> We agree this is a fair caveat: Table 1 is a broad SOTA comparison rather than a fully controlled fairness study. We would like to clarify this point in three parts:
>
> (1) In Table 1, VidLaDA, LLaDA-V, and SDAR-VL all use SigLIP-2, yet their performance differs widely. If SigLIP-2 alone were decisive, these models should be much closer. Also, on the LLM side, VidLaDA is built on the LLaDA backbone, while LLaVA-OneVision / LLaVA-Video use the stronger Qwen2-family backbones, so the comparison is not obviously biased in our favor there either.
>
> (2) Appendix F.4 provides a cleaner control: under identical data, hyperparameters, frozen ViT encoders, and schedule, AR and DLM baselines start from comparable performance. This reduces training-recipe confounds without fully removing backbone differences. To further reduce bias from incomplete fitting, we extended this matched setting to 4 epochs:
>
> |Model|#F|LongVideoBench|LVBench|MVBench|Video-MME|Avg|
> |---|---|---|---|---|---|---|
> |LLaVA-OV|32|56.5|40.7|56.7|58.5|53.1|
> |+SFT 1 Epoch|32|57.4 (+0.9)|38.8 (-1.9)|54.1 (-2.6)|58.6 (+0.1)|52.2 (-0.9)|
> |+SFT 4 Epoch|32|56.0 (-0.5)|38.3 (-2.4)|54.5 (-2.2)|56.8 (-1.7)|51.4 (-1.7)|
> |LLaDA-V|32|58.2|36.4|53.1|56.4|51.0|
> |+SFT 1 Epoch|32|58.9 (+0.7)|36.6 (+0.2)|53.7 (+0.6)|58.5 (+2.1)|51.9 (+0.9)|
> |+SFT 4 Epoch|32|58.0 (-0.2)|37.1 (+0.7)|53.6 (+0.5)|58.7 (+2.3)|51.9 (+0.9)|
>
> Both models keep their ViT encoders frozen. AR degrades at 4 epochs (52.2 vs 51.4), while DLM stays stable (51.9 vs 51.9), with Video-MME widening in DLM's favor (58.7 vs 56.8).
>
> (3) More broadly, Table 1 is SOTA context where cross-encoder comparison is standard in MLLM evaluation; our main fairness evidence remains the matched control in (2).
>
>
> > *Q2 - Persistent attention to early tokens despite bidirectional attention.*
>
> We would like to clarify that Global Anchor Tokens are fundamentally different from the "attention sink" phenomenon reported in StreamingLLM. As analyzed in Section 3.1, bidirectional DLMs have symmetric receptive fields and avoid the attention concentration induced by causal mask in AR decoding. Thus, the pattern in VidLaDA should not be interpreted as the same left-prefix artifact.
>
> Instead, the anchor pattern in VidLaDA arises from a different mechanism and is specifically associated with the visual modality. We interpret these anchors as stable cross-frame positions that carry temporally summarized information, rather than as a pseudo-pattern caused by causal left-prefix effects.
>
> More specifically, the reviewer's observed concentration on the first few tokens is more plausibly explained by a mismatch between text and visual attention magnitudes in Figure 3, rather than by AR-style positional bias. This is also consistent with the broader MLLM literature, where text tokens often receive disproportionate attention [R1]. In the standard setting (`system prompt + vision tokens + user prompt`), the first few high-attention positions in Figure 3 may therefore correspond to early text tokens, especially the system prompt. This is only a complementary explanation and does not replace our main visual-anchor interpretation: the Global Anchor Tokens used by MARS-Cache are still visual anchors. More importantly, compared with AR models, VidLaDA distributes attention over visual tokens much more evenly across temporal positions, exactly as highlighted by Section 3.1.
>
> > *Q3 - Connection to DyToK's "temporal attention outliers" and potential cross-analysis.*
>
> Thank you for this suggestion. DyToK interprets temporal attention outliers in AR MLLMs as attention artifacts and suppresses them with token-budget constraints.
>
> We believe our Global Anchor Token finding offers a complementary bidirectional view. Because the model structure is different, a fully bidirectional DLM can restore all-to-all interactions among video tokens and between the user input and the video input (Figure 1). Because bidirectional attention restores full connectivity, certain visual tokens can aggregate context information more effectively and therefore naturally emerge as globally informative anchors, which is difficult to expose under causal attention.
>
> Accordingly, the two methods take different actions on these high-attention positions: DyToK treats them as noise and suppresses them with token-budget constraints, whereas our MARS-Cache (Adaptive Anchor Token Searching, Section 4.2) explicitly identifies and preserves them to maintain global information flow under sparse attention. Appendix F.5.1 / Table 8 shows that removing anchors degrades accuracy. We will add this analysis in the camera-ready version.
>
> [R1] When Language Overrules: Revealing Text Dominance in Multimodal Large Language Models. arXiv, 2025.

---

> > ### Author Rebuttal · Reviewer_ajbR · 2026-04-04
> >
> > I sincerely appreciate the authors' detailed responses. All of my concerns have been adequately addressed. I hope the authors will incorporate our discussion into the final revision. Based on this, I maintain my positive evaluation.

---

> > > ### Author Response · Authors · 2026-04-04
> > >
> > > Thank you for your thoughtful review and for confirming that your concerns have been adequately addressed. We are glad our responses resolved your concerns. Following your advice, we will incorporate our discussion into the final revision. We appreciate your decision to maintain your positive evaluation. Thank you once again for your valuable insights and constructive suggestions, which have been instrumental in improving the quality of our work.

---

### Official Review · Reviewer_B97v · 2026-03-15

**Soundness:** 2
**Presentation:** 2
**Significance:** 2
**Originality:** 2
**Overall Recommendation:** 3
**Confidence:** 3

**Summary:**

This paper proposes VidLaDA, whose core idea is to shift video large models from the common AR decoding to DLM-style bidirectional decoding. The paper argues that key information in videos is distributed across space and time, and is not naturally suited for left-to-right causal modeling; AR's causal attention introduces asymmetric receptive fields, making earlier tokens easier to leverage while important visual evidence in the middle and later portions is more likely to be weakened. To address this, the authors employ bidirectional attention for video understanding and combine it with parallel denoising to achieve higher generation throughput.

**Compliance With Llm Reviewing Policy:**

Affirmed.

**Key Questions For Authors:**

See Weaknesses

**Limitations:**

Yes

**Strengths And Weaknesses:**

Strengths：
1.VidLaDA proposes a DLM video model integrated with MARS-Cache. It analyzes why bidirectional attention is suitable for video.
2.Experiments cover multiple public video understanding benchmarks, and the results show that VidLaDA consistently outperforms the DLM baseline, while demonstrating competitive performance against AR models on long video and spatiotemporal reasoning-related tasks.
Weakness:
1.Table 1 shows that different models use significantly varying numbers of input frames. This suggests that the performance improvements shown in the main table may be partially attributed to differences in frame count rather than model architecture alone. Could you provide an ablation study for VidLaDA examining its performance across different frames?
2.It would be valuable to verify whether MARS-Cache maintains its throughput advantages under conditions beyond the reported settings. Additionally, please include ablation studies on MARS-Cache's key hyperparameters to demonstrate the robustness and sensitivity of the proposed method.

---

> ### Author Rebuttal · Authors · 2026-03-30
>
> Thank you for your valuable feedback. We provide detailed responses below. The reviewer's remarks are in *italics*, followed by our responses.
>
>
>
> > *W1 - Frame count ablation for VidLaDA across different numbers of input frames.*
>
>
>
> We provide VidLaDA's performance across 8–64 frames alongside LLaVA-OneVision (LLaVA-OV) at 32 frames, covering all Table 1 benchmarks:
>
> | Model    | #F   | Video-MMMU | LongVideoBench | LVBench | EgoSchema | MVBench | MLVU_dev | MLVU_test | Video-MME |
> | -------- | ---- | ---------- | -------------- | ------- | --------- | ------- | -------- | --------- | --------- |
> | LLaVA-OV | 32   | 33.9       | 56.5           | 40.7    | 60.1      | 56.7    | 64.7     | 45.3      | 58.5      |
> | VidLaDA  | 8    | 44.8       | 58.6           | 38.9    | 61.1      | 58.0    | 57.6     | 39.5      | 55.0      |
> | VidLaDA  | 16   | 46.1       | 60.1           | 40.4    | 62.6      | 59.2    | 61.4     | 45.3      | 57.5      |
> | VidLaDA  | 32   | 47.1       | 59.3           | 41.7    | 63.8      | 59.1    | 65.1     | 49.9      | 62.1      |
> | VidLaDA  | 64   | 46.6       | 61.4           | 44.7    | 64.5      | 59.4    | 69.2     | 53.4      | 64.1      |
>
> VidLaDA at 16 frames already matches or exceeds LLaVA-OV at 32 frames on most benchmarks. At 32 frames, VidLaDA outperforms LLaVA-OV across all benchmarks. These results suggest that the gains are not explained solely by frame count differences, consistent with the analysis in Section 3.1.
>
>
>
> > *W2 - MARS-Cache robustness beyond reported settings and hyperparameter sensitivity ablations.*
>
>
>
> Beyond the paper's 32-frame Table 2 setting, we additionally evaluate all decoding variants at 8/16/32 frames (Acc / TPS):
>
> | **Model**    | **#F** | **Ego_subset** | **LongVideoBench** | **MLVU_test** |
> | ------------ | ------ | -------------- | ------------------ | ------------- |
> | LLaVA-OV     | 8      | 63.0 / 41.4    | 54.1 / 37.7        | 37.9 / 40.2   |
> | VidLaDA      | 8      | 63.8 / 14.4    | 58.4 / 19.9        | 38.7 / 15.0   |
> | VidLaDA+Dual | 8      | 64.6 / 33.6    | 58.0 / 39.0        | 37.3 / 26.1   |
> | VidLaDA+MARS | 8      | 64.6 / 60.3    | 58.5 / 81.7        | 38.3 / 71.6   |
> | LLaVA-OV     | 16     | 62.6 / 37.8    | 56.1 / 32.5        | 39.1 / 35.3   |
> | VidLaDA      | 16     | 68.2 / 6.5     | 60.7 / 8.8         | 42.9 / 7.8    |
> | VidLaDA+Dual | 16     | 64.4 / 22.1    | 60.2 / 24.2        | 42.4 / 19.6   |
> | VidLaDA+MARS | 16     | 65.6 / 46.9    | 59.9 / 62.4        | 41.9 / 56.3   |
> | LLaVA-OV     | 32     | 62.8 / 27.0    | 57.5 / 25.0        | 43.3 / 27.2   |
> | VidLaDA      | 32     | 67.4 / 2.7     | 59.8 / 2.1         | 50.2 / 3.3    |
> | VidLaDA+Dual | 32     | 67.2 / 23.7    | 59.4 / 18.5        | 50.5 / 26.0   |
> | VidLaDA+MARS | 32     | 67.0 / 33.6    | 59.7 / 25.2        | 50.7 / 33.6   |
>
> MARS-Cache maintains a clear TPS advantage on these additional settings while remaining competitive in accuracy. Beyond reported settings, MARS-Cache provides an effective accuracy–throughput trade-off compared to VidLaDA without caching (highest accuracy but lowest TPS at most settings).
>
> Separately, regarding robustness and sensitivity, these additional results are not based on an extensive benchmark-specific hyperparameter search for MARS-Cache. Together with the ablations below, they suggest that the method is not overly sensitive to a single operating point. To further verify the robustness and sensitivity of the design, we additionally evaluate the same hyperparameter designs as in Tables 8-10 at 8, 16, and 32 frames, and summarize the gap between the paper setting and the best configuration on EgoSchema (subset) / MLVU (test) below.
>
> | **Ablation**        | **8F**    | **16F**   | **32F**   |
> | ------------------- | --------- | --------- | --------- |
> | Anchor Count (T8)   | 0.0 / 0.0 | 3.2 / 1.5 | 0.0 / 0.0 |
> | Layer Refresh (T9)  | 3.0 / 0.0 | 1.4 / 0.9 | 0.0 / 0.5 |
> | Search Tokens (T10) | 1.2 / 0.0 | 1.2 / 1.5 | 0.0 / 0.6 |
>
> The 32-frame setting is always within 0.6 points of the best variant, and even at 8/16 frames, the gaps remain moderate. We therefore view the MARS design as reasonably robust across frame counts rather than narrowly tuned to one operating point.

---

### Decision · Program_Chairs · 2026-04-30

**Decision:**

Accept (regular)

**Comment:**

This paper proposes VidLaDA, a Video LLM that replaces standard autoregressive decoding with a bidirectional Diffusion Language Model approach and utilizes a novel MARS-Cache strategy to mitigate computational overhead. The paper received final scores of (3, 5, 4, 5). The reviewers acknowledged the creative departure from the "causal mask" limitations, and the authors successfully resolved some raised concerns regarding memory usage, training costs, and bidirectional isolation. Since there are no remaining major concerns after the rebuttal, the AC recommends accept.